# TRUSTSQL: BENCHMARKING TEXT-TO-SQL RELIABILITY WITH PENALTY-BASED SCORING

## ABSTRACT

Text-to-SQL enables users to interact with databases using natural language, simplifying information retrieval. However, its widespread adoption remains limited for two main reasons: (1) existing benchmarks focus solely on feasible questions that can always be mapped to SQL queries, overlooking infeasible questions that cannot, and (2) current models lack abstention mechanisms, posing the risk of providing incorrect answers. To address these gaps, we introduce TrustSQL, a new benchmark designed to evaluate text-to-SQL reliability. At its core is the proposed Reliability Score (RS), which quantifies a model's helpfulness (correct answers) relative to its harmfulness (incorrect answers weighted by a user-defined penalty). TrustSQL is constructed by re-annotating three datasets—ATIS, Advising, and EHRSQL—while incorporating infeasible questions to enable comprehensive evaluations across diverse model inputs. We evaluate text-to-SQL models integrated with various abstention mechanisms, leveraging classification and uncertainty estimation methods. Our experiments reveal that only a few models achieve positive scores (*i.e.*, helpfulness outweighing harmfulness) under high-penalty settings, indicating that most models are unsuitable for deployment in safety-critical scenarios. This underscores the need to develop models that not only improve SQL generation but also guarantee a certain degree of reliability, ensuring safe deployment.

## 1 INTRODUCTION

Relational databases store vast amounts of information and are fundamental to decision-making in various domains such as healthcare (Hillestad et al., 2005), finance (Beck et al., 2000), and e-commerce (Akter & Wamba, 2016). Text-to-SQL, the task of converting natural language questions into SQL queries, has emerged as a powerful tool for enabling database question-answering (DBQA) systems (Androutsopoulos et al., 1995; Zhong et al., 2017; Yu et al., 2018; Li et al., 2023). However, deploying such models in real-world settings requires assessing both the potential helpfulness and harmfulness they may introduce[1], in line with specific user safety requirements. The *helpfulness* of these models lies in their ability to handle input questions accurately—either by providing correct answers to questions that have ground-truth (GT) answers or by abstaining from answering questions with no GT answer (Rajpurkar et al., 2018). Conversely, *harmfulness* arises from inaccurate handling—either by providing incorrect answers to questions that have GT answers or by failing to abstain from answering questions that have no GT answer. Given this context, we consider a model more *reliable* if the difference between its helpfulness and harmfulness exceeds that of other models, making it more suitable for deployment.

Current text-to-SQL benchmarks, however, fall short in assessing these aspects for two main reasons: **(1) Limited Scope of Questions:** Most text-to-SQL models and benchmarks focus on questions that can be directly mapped to SQL queries (*i.e.*, *feasible* questions). However, this limited scope overlooks the variety of questions users may pose in real-world scenarios (see Figure 1a). For example, questions that reference non-existent columns (*e.g.*, "Which physician treated patient 1001?" when the physician column is missing) or contain ambiguous phrasing (*e.g.*, "What's the

---

[1]The model's output, whether helpful or harmful, depends on the user's decision, making it 'potential' in the context of model evaluation. For readability, we may omit the term 'potential' moving forward.

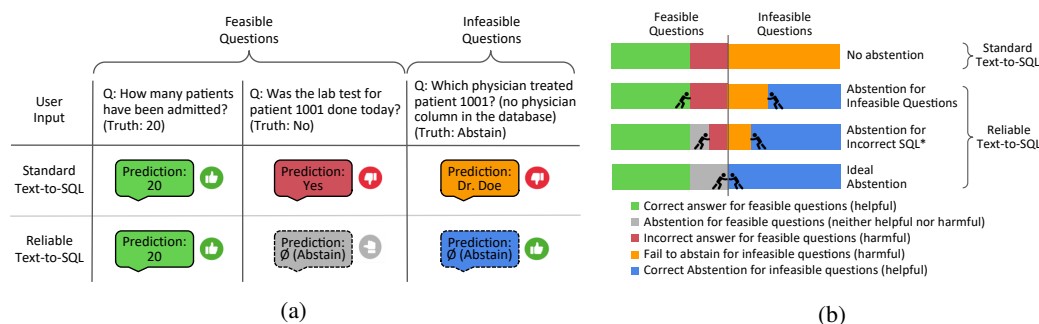

(a)                                                                                          (b)

Figure 1: **(a) Extended Question Scope.** Standard text-to-SQL models respond to every user question without considering the possibility of infeasible questions. On the other hand, reliable text-to-SQL models (this work) handle any user inputs and avoid responses that could lead to potential harm. **(b) Incorporating Abstention.** Reliable text-to-SQL models should not only abstain from providing incorrect answers to feasible questions, but also to infeasible ones. Even if the model cannot answer all feasible questions correctly, users will trust it if it consistently provides correct answers to the questions it can confidently answer. *Incorrect SQL includes incorrect SQL generation for both feasible and infeasible questions (any attempt to generate SQL is also incorrect for infeasible questions).

score for EECS 545?" when multiple score-related columns exist) do not have GT answers and require further clarification to be properly addressed (*i.e.*, *infeasible* questions) (Zhang et al., 2020; Wang et al., 2023a). Consequently, even high-performing models on benchmarks like Spider (Yu et al., 2018) and BIRD (Li et al., 2023) may require significant adjustments to be deployed as DBQA systems to handle these cases. **(2) Lack of Abstention:** Standard text-to-SQL models attempt to generate SQL queries for all input questions, lacking the ability to abstain when faced with questions that are likely to result in incorrect answers or are intrinsically infeasible (see Figure 1b). Without abstention mechanisms or other safety measures, they may continue to risk providing incorrect answers, potentially undermining users' trust in these models. (Whitehead et al., 2022; Lee et al., 2022; Chen et al., 2023).

To address these shortcomings, we introduce *TrustSQL*, a new benchmark designed to evaluate the reliability of text-to-SQL models. *Text-to-SQL reliability* is quantified by our new metric, the *Reliability Score* (RS), which assigns a score of $+1$ when the model accurately handles input questions and $-c$ when it does not, where the penalty $c$ reflects the user safety requirements. Alongside the evaluation scheme, we re-annotate three complex, domain-specific text-to-SQL datasets—ATIS (Hemphill et al., 1990), Advising (Finegan-Dollak et al., 2018), and EHRSQL (Lee et al., 2022)—each derived from real user utterances within their respective domains (details in Section 4.1), creating a high-quality benchmark for reliability. We also incorporate three common categories of infeasible questions—`missing-schema`, `ambiguous`, and `non-sql` (details in Section 4.3)—to enable a more comprehensive and realistic evaluation of model performance when handling diverse user inputs.

To evaluate text-to-SQL models for this benchmark, we implemented various models featuring compatible abstention mechanisms, spanning model architectures (decoder-only and encoder-decoder) and training methods (fine-tuning and in-context learning) to enhance reliability. Experimental results reveal that, while high-performing models such as GPT-4o-powered baselines excel in low-penalty settings, models utilizing adjustable thresholds and uncertainty estimation outperform them in high-penalty settings. Notably, only a handful of models achieve a positive RS under the highest penalty settings, revealing that most models tend to be more harmful than helpful and underscoring the need to develop models suitable for deployment in safety-critical scenarios.

## 2 RELATED WORKS

**Text-to-SQL Benchmarks** Existing benchmarks like TriageSQL (Zhang et al., 2020) and DTE (Wang et al., 2023a) include both feasible (answerable) and infeasible (unanswerable and ambigu-

Table 1: Comparison of TrustSQL with existing benchmarks. [∧] Indicates whether the dataset contains question-SQL pairs and corresponding databases that can produce execution results. [§] Indicates whether the dataset contains textual hints/knowledge to guide SQL generation. [†] Indicates that the dataset contains string-formatted database schemas specific to each sample (TriageSQL) or single-table settings (DTE and WikiSQL). [‡] Indicates the number of unique tables used for each SQL query. [*] Indicates whether model mistakes are penalized during evaluation.

| Dataset | UnansQ | AmbigQ | AnsQ | ExecSQL[∧] | Evidence[§] | #DB | #Table/DB | #Table/SQL[‡] | Avg SQL Tok | Eval Penalty[*] |
|---|---|---|---|---|---|---|---|---|---|---|
| TriageSQL | ✓ | ✓ | ✓ | ✗ | ✗ | -[†] | 1.7 | - | - | ✗ |
| DTE | ✓ | ✓ | ✓ | ✗ | ✗ | - | 1.0 | - | - | ✗ |
| WikiSQL | ✗ | ✗ | ✓ | ✓ | ✗ | - | 1.0 | 1.0 | 12.1 | ✗ |
| Spider | ✗ | ✗ | ✓ | ✓ | ✗ | 200 | 5.1 | 1.6 | 18.6 | ✗ |
| KaggleDBQA | ✗ | ✗ | ✓ | ✓ | ✗ | 8 | 2.3 | 1.2 | 17.3 | ✗ |
| EHRSQL | ✓ | ✗ | ✓ | ✓ | ✗ | 1 | 17.0 | 2.4 | 68.9 | ✗ |
| BIRD | ✗ | ✗ | ✓ | ✓ | ✓ | 95 | 7.5 | 2.0 | 31.1 | ✗ |
| TrustSQL | ✓ | ✓ | ✓ | ✓ | ✓ | 3 | 18.0 | 2.9 | 60.5 | ✓ |

ous) questions, enabling question classification prior to SQL generation. However, they lack executable SQL and corresponding databases, which limits their scope to address SQL generation errors. Standard text-to-SQL benchmarks, such as WikiSQL (Zhong et al., 2017), Spider (Yu et al., 2018), KaggleDBQA (Lee et al., 2021), and BIRD (Li et al., 2023), focus on increasing the difficulty of question-to-SQL mapping by enhancing SQL complexity and database schemas, but they assume that all input questions are feasible. This leaves the challenge of handling infeasible questions unaddressed and overlooks the importance of abstaining when SQL generation is likely to fail, as their evaluation schemes do not penalize incorrect answers—an essential consideration for reliable system deployment. EHRSQL (Lee et al., 2022) incorporates both answerable and unanswerable questions into the SQL generation task based on a hospital user survey. However, it lacks a dedicated reliability metric for accounting for diverse user safety requirements, and its template-based generation of infeasible questions limits diversity (Yang et al., 2024). To address these gaps, TrustSQL employs manually annotated infeasible questions that closely resemble feasible ones, introduces a new metric aligned with user-defined safety criteria, incorporates penalties for incorrect answers in its evaluation, and leverages datasets from three distinct domains to evaluate whether methodological trends hold consistently across databases. Table 1 summarizes how TrustSQL compares to other text-to-SQL benchmarks.

**Uncertainty Estimation in NLP** Uncertainty estimation enhances AI system safety by quantifying model confidence and identifying potential errors (Hendrycks et al., 2021). In natural language processing (NLP), uncertainty estimation methods can be categorized into four broad approaches: (1) Probability-based methods, which quantify uncertainty through probabilistic outputs, such as entropy or maximum softmax probabilities, providing a measure of confidence in model predictions (Xiao & Wang, 2021; Kim et al., 2022; Stengel-Eskin & Van Durme, 2023); (2) Distance-based methods, which assess uncertainty by calculating the deviation of new inputs from the training data distribution, often using metrics like the Mahalanobis distance between features to identify outliers (Lee et al., 2018; Ren et al., 2021; 2022); (3) Sampling-based methods, which utilize techniques such as Monte Carlo Dropout or Self-Consistency to generate multiple predictions for the same input, quantifying uncertainty through the variance in model outputs (Gal & Ghahramani, 2016; Wang et al., 2022); and (4) Text-based methods, which employ language models to generate both task outputs and their corresponding confidence levels in natural language (Lin et al., 2022; Kadavath et al., 2022). While significant research has focused on quantifying uncertainty, its application to text-to-SQL has been less explored due to the absence of benchmarks addressing safety and model abstention. TrustSQL aims to bridge this gap by providing a concrete benchmark, enabling the exploration and evaluation of various uncertainty estimation methods applied to text-to-SQL.

## 3 PROBLEM DEFINITION

Evaluating the reliability of text-to-SQL models is essential to ensure their safe and effective use in real-world DBQA systems. These models can be potentially helpful or harmful, depending on

the correctness of their output in different contexts. A system is considered *helpful* when it makes *correct decisions*. This includes generating correct SQL queries for feasible questions (denoted as $Q$), returning results matching the ground-truth (GT) SQL query (denoted as $S$), or abstaining from answering infeasible questions (denoted as $Q^c$) where no valid SQL query exists (represented by $\phi$). Conversely, a system becomes *harmful* when it makes *incorrect decisions*, such as generating incorrect SQL for feasible questions or attempting to answer infeasible questions where SQL generation is guaranteed to be incorrect.

Given these criteria for helpful and harmful behaviors, we need a quantitative measure to evaluate model reliability to answer if it is deployable or not. To this end, we introduce the *Reliability Score* (RS). The RS measures the difference between a model's helpfulness (defined as the number of correct decisions) and its harmfulness (defined as the number of incorrect decisions, weighted by a penalty factor $c$). The penalty factor $c$ reflects how many correct decisions are needed to offset one incorrect decision. The RS is calculated as follows:

$$\Phi_c(x,y) = \begin{cases} 1 & \text{if } \{x,y\} \in D_Q, g(x) = 1, Acc(f(x), y) = 1 \\ 0 & \text{if } \{x,y\} \in D_Q, g(x) = 0 \\ -c & \text{if } \{x,y\} \in D_Q, g(x) = 1, Acc(f(x), y) = 0 \\ -c & \text{if } \{x,y\} \in D_{Q^c}, g(x) = 1 \\ 1 & \text{if } \{x,y\} \in D_{Q^c}, g(x) = 0, \end{cases} \tag{1}$$

where $D_Q$ is the set of feasible questions with GT SQL queries $(Q, S)$; $D_{Q^c}$ is the set of infeasible questions with no GT SQL $(Q^c, \phi)$; $f(x)$ is the model's generated SQL query for input $x$; $Acc(f(x), y)$ represents execution accuracy (*i.e.*, whether the answer from the generated SQL matches that of the GT query); $g(x)$ indicates the model's decision to provide an answer or abstain (1 for answering, 0 for abstaining); and $c$ is a user-defined penalty that adjusts the significance of incorrect decisions. The total score $\Phi_c$ is the mean over all $N$ individual scores for each sample[2].

**Model selection with RS**  A model can be considered potentially more harmful than helpful depending on the contexts, as reflected in the penalty term $c$. In fact, the choice of the penalty $c$ is user-defined and depends on the safety requirements for model deployment, varying possibly based on user preferences, SQL proficiency, or organizational policies. We provide a few practical starting points to set the penalty and model selection guidelines:

- **Lenient scenario** ($c = 1$): A single mistake carries the same weight as one correct decision. A positive RS here indicates the model makes more correct decisions than incorrect ones.
- **Moderate scenario** ($c = 10$): Every 10 correct decisions carry the same weight as one incorrect decision, reflecting a moderate tolerance for errors.
- **Strict scenario** ($c = N/2$): Two mistakes result in a negative RS, even if all other decisions ($N{-}2$ out of $N$ total cases) are correct. This scenario emphasizes high reliability.
- **Most strict scenario** ($c = N$): A single mistake results in a negative RS, even if all other decisions ($N{-}1$ cases) are correct. Models must avoid any mistakes to achieve a positive RS.

Once the safety requirement for model deployment is set, the best model can be selected based on its RS performance. Below are key guidelines for model selection using the RS:

- **Model comparison**: A model's superior performance under a specific penalty does not imply it outperforms others across different penalty settings. Models should be compared within the same penalty setting.
- **Interpreting the score**: Models with positive RS values are preferable for deployment, as they meet safety requirements and demonstrate greater helpfulness than harmfulness. A model that ranks the highest but achieves a negative RS should be reconsidered for deployment

# 4  DATA CONSTRUCTION

A DBQA system's inputs can be classified as either feasible or infeasible. Since benchmark datasets cannot capture every possible variation of these questions, we design TrustSQL to reflect common

---

[2]The upper bound of the RS is 100% and the lower bound is $-c \times 100\%$ .

scenarios that systems are likely to encounter. Specifically, we include both feasible questions that reflect real user needs and infeasible questions derived from common failure cases observed in user studies.

## 4.1 DATABASE SELECTION

To construct TrustSQL, we first select databases based on two main criteria: (1) the presence of complex schemas that mirror the intricacies of real-world databases, and (2) the availability of associated question intents that reflect genuine user intents regarding their complex database. Based on these criteria, we choose three domain-specific text-to-SQL datasets: ATIS (airline travel) (Hemphill et al., 1990), Advising (education) (Finegan-Dollak et al., 2018), and EHRSQL (healthcare) (Lee et al., 2022). These datasets feature complex database schemas (ATIS: 25 tables; Advising: 12 tables; EHRSQL: 17 tables) and capture real user needs that a DBQA system is likely to encounter.

We focus on these domain-specific datasets rather than including additional databases and question-SQL pairs from large-scale, cross-domain datasets like Spider (Yu et al., 2018) or BIRD (Li et al., 2023) for two main reasons: First, the nature of our task and the penalty-based scoring—which amplifies the impact of incorrect model decisions—necessitates high-quality annotations. Template-free SQL annotations obtained through crowdsourcing cannot be scalably maintained in terms of data quality. Second, these large-scale datasets have other limitations for inclusion in our task. For example, Spider is largely solved by models like GPT-4o, with remaining errors often due to annotation issues rather than challenges in SQL generation. BIRD relies on 'sample-level' evidence, which is incompatible with TrustSQL's setup where not all input questions are text-to-SQL feasible. TrustSQL uses 'database-level' evidence (*i.e.*, SQL assumption text in Appendices A.1.1, A.1.2, and A.1.3) shared across samples, removing the assumption that all input questions are feasible while still enabling the text-to-SQL task by using evidence only when necessary.

## 4.2 FEASIBLE QUESTIONS

Constructing high-quality data is critical for accurately evaluating the reliability of text-to-SQL models in TrustSQL. To ensure the highest quality of annotations, the three authors (all proficient in SQL) serve as annotators for creating the benchmark data, without relying on crowdsourcing. The annotation process involves the following steps: **Template Review and Modification**. One annotator thoroughly reviews all question templates and SQL structure annotations across the three datasets (one question template corresponds to one unique SQL structure). Templates that are semantically identical are merged, resulting in approximately 550 templates. **Paraphrase Generation**. Two annotators generate new natural language questions for each question template using GPT-4o and review them to ensure they coherently reflect the original question template. Additionally, the annotators check whether the paraphrases contain any implicit assumptions related to their corresponding SQL structures and draft a SQL assumption text if dataset-level assumptions exist. **Pair Construction**. One annotator merges the paraphrases with the question templates. This process involves inserting values into both the question paraphrases and the SQL queries derived from the SQL structures. **Review and Quality Control**. After merging question templates and paraphrases, all annotators engage in real-time discussions to resolve any disagreements regarding the created question-SQL pairs and SQL assumption text (Appendices A.1.1, A.1.2, and A.1.3). Discussions continue until a consensus is reached. Further details are provided in Appendix A.2.

## 4.3 INFEASIBLE QUESTIONS

We identify three common categories of infeasible questions from real user studies conducted in EHRSQL (Lee et al., 2022) and DTE (Wang et al., 2023a): (1) questions that cannot be answered based on existing database columns (`missing-schema`), (2) questions that lack sufficient information and require further interaction with the user to generate correct SQL queries[3] (`ambiguous`), and (3) questions that require operations beyond SQL's scope (`non-sql`).

Based on these findings, we manually annotate infeasible questions across all three databases. The manual annotation process involves the following steps: **Keyword-Guided Annotation**. Annota-

---

[3]We also assume single-turn text-to-SQL (*i.e.*, one user question followed by one model response). As a result, questions requiring clarification are considered infeasible.

Table 2: Sample questions in the ATIS portion of TrustSQL. The 'Note' column indicates question categories associated with each infeasible question.

| Type | Question | Note |
|---|---|---|
| Feasible | List all the flights that arrive at MKE airport. | - |
| Infeasible | Find the average number of lounges across all airports. | `missing-schema`
(No lounge information in DB) |
| | Which cities have the best airport ground services available? | `ambiguous`
("Best" is not defined) |
| | How can aircraft be clustered based on their range and cruising speed? | `non-sql`
(Clustering is not supported) |

tors are given specific keywords for annotating infeasible questions (*i.e.*, hypothetical columns for `missing-schema` questions, ambiguous terms for `ambiguous` questions, and task names for `non-sql` questions), along with a list of sample feasible questions. They are tasked to manu-ally modify the feasible questions using the provided keywords to make them infeasible—creating questions that resemble feasible ones but are, in fact, infeasible. For example, given the keyword 'appointment' in a hospital database where there is no record of next visits (`missing-schema`), a feasible question like "Has patient 0000 gotten any medication this year?" can be modified to "Has patient 0000 gotten any medication this year and has any next appointment?", which is no longer feasible given the database schema. **Review and Quality Control**. The questions are re-viewed in real-time by the annotators until all disagreements are resolved to ensure consistency and high-quality annotations. Common disagreements include: the phrasing of the annotated question not explicitly referencing the given hypothetical column or intended non-SQL tasks (*e.g.*, requesting clustering instead of grouping—the question phrasing must explicitly specify "clustering" to avoid confusion with grouping which exists in SQL), and questions using SQL but could technically be addressed with SQL (*e.g.*, conducting data normalization by calculating the mean and variance). More details about these keywords and the advantages of using keyword-based over template-based data creation (Lee et al., 2022) are discussed in Appendices A.3 and A.4.

## 4.4 DATA OVERVIEW

This section provides an overview of data in TrustSQL, detailing the distribution of feasible and infeasible questions across different data splits.

**Feasible Data**    Text-to-SQL model performance is influenced by several factors, including the complexity of SQL queries (query difficulty) (Yu et al., 2018) and the presence of the same SQL structures (but with differently paraphrased questions) in the training set (query familiarity) (Finegan-Dollak et al., 2018). To better understand these factors, we tag each sample for both diffi-culty and familiarity. **Query difficulty.** We classify SQL queries into three categories based on their complexity, following the criteria introduced in DIN-SQL (Pourreza & Rafiei, 2023): `easy` (single-table queries with no joins, nesting, or set operations), `medium` (queries with joins but no nesting or set operations), and `hard` (queries with joins, subqueries, or set operations). **Query familiarity.** First, we split the data into training, validation, and test set evenly based on SQL structures, with each set containing disjoint structures. Since each structure is associated with multiple questions, we sample data from the training set without replacement and allocate it to both the validation and test sets. We refer to these sampled questions as *familiar*, because their corresponding structures can be available to the model during fine-tuning or few-shot prompting. The questions whose structures are not present in the training set are referred to as *unfamiliar*. Since feasible data questions are created in a template-based manner, this process is possible and enables maintaining control over the semantic distribution of questions across data splits.

**Infeasible Data**    Infeasible questions are designed to simulate real-world text-to-SQL deployment scenarios where user questions often do not correspond to any valid SQL queries. To ensure balanced evaluation, we set the number of infeasible questions to match the number of feasible ones in the test set. In task-oriented dialog systems, out-of-scope intents (*i.e.*, infeasible questions) are deliberately excluded from the training set Zheng et al. (2020); Marek et al. (2021). Following this practice, we include these questions only in the test set, reflecting unexpected encounters during deployment.

Table 3: Dataset statistics of TrustSQL. TrustSQL includes an equal number of feasible and infeasible questions in each test set. All samples are human-verified.

|  | ATIS | Advising | EHRSQL |
|---|---|---|---|
| Train | 623 | 698 | 2633 |
| Valid | 407 | 471 | 559 |
| Test | 794 | 918 | 1118 |
| ↪ Feasible | 397 | 459 | 559 |
| ↪ Infeasible | 397 | 459 | 559 |
| Total | 1824 | 2067 | 4310 |

**Overall Statistics**   Table 3 summarizes the data distribution across all splits. Detailed statistics on query difficulty are provided in Appendix A.

## 5 EXPERIMENTS

In this section, we introduce various abstention mechanisms integrated with text-to-SQL models, along with the RS and other evaluation metrics.

### 5.1 MODELS

To illustrate how reliable models can be constructed to tackle this benchmark, we implement text-to-SQL models using both fine-tuning (FT) and in-context learning (ICL) with different model architectures. Each architecture is integrated with two distinct styles of abstention mechanisms—Classifier-based and Uncertainty-estimation-based—depending on their compatibility. The implementation details are reported in Appendix C. Note that we treat any SQL generation that lacks a SELECT clause as an abstention, even if no explicit abstention mechanisms are in place.

- **Classifier (CL)-based**: These models operate sequentially, with task-specific sub-models functioning in the following order: infeasible question detection, SQL generation, and SQL error detection. Each sub-model is designed to be plug-and-play, offering high interpretability but also posing the risk of error propagation between sub-models.
- **Uncertainty-estimation (UE)-based**: These models use a single model framework to determine abstention decisions based on uncertainty estimation, resulting in a more streamlined implementation. However, this simplicity comes at the cost of interpretability and control, as the SQL generation and abstention mechanisms are closely integrated.

#### 5.1.1 FINE-TUNING

We use two backbone models with different architectures for SQL generation: SQLCoder 7B-2 (decoder-only, SQL-specialized Code Llama) and T5-3B (encoder-decoder). Each model employs abstention mechanisms tailored to its architecture.

**CL-Based**   When using SQLCoder-2 as a SQL generator (denoted as SQLCODER2), we develop two additional classifiers: an infeasible question detector and a SQL error detector. The infeasible question detector (+INF†) is fine-tuned on the training set from the classification dataset TriageSQL[4] The SQL error detector (+ERR†) is fine-tuned on the accuracy labels from the validation set of SQLCODER2.

**UE-Based**   To leverage T5's encoder-decoder architecture (denoted as T5-3B), we use four uncertainty estimation methods for abstention: (1) entropy-based, assigning the max entropy value from generated SQL tokens (+MAXENT (Lee et al., 2022)); (2) probability-based, using the minimum of the highest token probabilities in the sequence (+MAXPROB (Stengel-Eskin & Van Durme, 2023)); (3) Mahalanobis distance from training data features (+FEATMD (Ren et al., 2022)); and (4) relative Mahalanobis distance from a background distribution in the TriageSQL dataset (+FEATRMD (Ren et al., 2022)). Also, these methods require calibrated thresholds to distinguish in-distribution from out-of-distribution (OOD) samples. As a result, we propose an automatic threshold selection

---

[4]There exist two openly available datasets for infeasible question classification. We use TriageSQL (Zhang et al., 2020) as it is the only one that covers multi-table settings.

method (Appendix C.1.2), treating correct SQL as in-distribution and incorrect SQL as OOD, based on validation set performance.

### 5.1.2 IN-CONTEXT LEARNING

We use GPT-4o for SQL generation (denoted as GPT-4O) through retrieval-based few-shot prompting, utilizing eight question-SQL pairs selected based on the maximal marginal relevance criterion (Carbonell & Goldstein, 1998). For comparison, we also present the performance of baseline SQL generators and other state-of-the-art cross-domain text-to-SQL methods, such as DIN-SQL (task decomposition) (Pourreza & Rafiei, 2023) and MAC-SQL (multi-agent framework) (Wang et al., 2023b), in Appendix B.2.

**CL-Based** For the infeasible question detector, we use chain-of-thought prompting, providing definitions for the three categories of infeasible questions without few-shot examples (+INF$^{\ddagger}$). This setup generates 'Feasible' or 'Infeasible' labels for the input questions. For SQL error detection, we adapt DIN-SQL's zero-shot self-correction prompt to generate 'Correct' or 'Incorrect' labels based on both the input question and the SQL output from GPT-4O (+ERR$^{\ddagger}$).

**UE-Based** We apply three abstention methods to GPT-4O: (1) +VERBAL, where eight correct and eight incorrect question-SQL pairs are provided, each labeled as either 'True' or 'False,' based on the validity of the question-SQL pairs. Abstention occurs when a 'False' label is generated following the SQL (Lin et al., 2022; Kadavath et al., 2022); (2) +VOTE-SQL, where GPT-4O is sampled five times for SQL generation, and the model abstains if the generated SQL outputs are not identical (Wang et al., 2022); and (3) +VOTE-ANS, similar to +VOTE-SQL, but abstention is triggered based on the consistency of the answers retrieved by the generated SQL queries.

### 5.2 EVALUATION METRICS

We apply three penalties to the RS: $\Phi_1$, $\Phi_{N/2}$, and $\Phi_N$. Additionally, we incorporate three metrics from the selective prediction literature (El-Yaniv et al., 2010), adapted to our framework. These metrics include: coverage on feasible questions ($\mathcal{C}_Q$), the proportion of feasible questions that the model chooses to answer; risk on feasible questions ($\mathcal{R}_Q$), the proportion of incorrect answers among the answered feasible questions; and risk on infeasible questions ($\mathcal{R}_{Q^c}$), the proportion of answered infeasible questions. These metrics reflect the model's performance in terms of the ratio of correctness, while the RS directly measures user-defined text-to-SQL reliability.

## 6 RESULTS

### 6.1 FULL DATA, $D_Q \cup D_{Q^c}$

Table 4 shows the performance of various text-to-SQL models with integrated abstention mechanisms across the three datasets.

**Coverage-Risk** Incorporating abstention significantly lowers $\mathcal{R}$ across all models, but at the cost of a reduction in $\mathcal{C}_Q$. Models with error detectors (+ERR$^{\dagger\ddagger}$) excel at identifying errors in SQL queries for feasible questions (*i.e.*, lower $\mathcal{R}_Q$), while infeasible question detectors (+INF$^{\dagger\ddagger}$) more effective at recognizing infeasible questions (*i.e.*, lower $\mathcal{R}_{Q^c}$), highlighting the complementary roles of these detectors. As a result, combining both detectors yields much better overall $\mathcal{R}$ reduction, though it further decreases $\mathcal{C}_Q$. Probability-based methods (+MAXENT and +MAXPROB) and distance-based (+FEATMD and +FEATRMD) methods offer more flexibility in threshold selection compared to text- or sampling-based methods. Stricter thresholds (Appendix C.1.2) in these models achieve very low $\mathcal{R}$ ($< 2.1\%$) but at the cost of substantially reducing $\mathcal{C}_Q$ ($< 11.3\%$). Meanwhile, text- and sampling-based methods show moderate improvements in both $\mathcal{C}_Q$ (52.9%) and $\mathcal{R}$ (8.0%). +VOTE-SQL is stricter than +VOTE-ANS because it does not allow different SQL structures as model outputs, but it more effectively reduces risk, particularly in $\mathcal{R}_{Q^c}$ (12.2% → 3.9%).

**Reliability** The CL-based FT model is competitive with the CL-based ICL model in $\Phi_1$ when both detectors (+INF$^{\dagger}$+ERR$^{\dagger}$) are included. However, the ICL model outperforms under higher penalty settings, as result of better reduction in $\mathcal{R}_Q$. Text-based and sampling-based methods achieve high

Table 4: Model comparison for the full data. For all metrics, higher values are better, except for $\mathcal{R}_Q$ and $\mathcal{R}_{Q^c}$. The scores in the shaded cells represent SQL generation performance without abstention. All metrics are in % and numbers in thousands are abbreviated as K for readability.

| | | ATIS | | | | | Advising | | | | | EHRSQL | | | | |
|---|---|---|---|---|---|---|---|---|---|---|---|---|---|---|---|---|
| | | $\mathcal{C}_Q$ | $\mathcal{R}_Q / \mathcal{R}_{Q^c}(\downarrow)$ | $\Phi_1$ | $\Phi_{N/2}$ | $\Phi_N$ | $\mathcal{C}_Q$ | $\mathcal{R}_Q / \mathcal{R}_{Q^c}(\downarrow)$ | $\Phi_1$ | $\Phi_{N/2}$ | $\Phi_N$ | $\mathcal{C}_Q$ | $\mathcal{R}_Q / \mathcal{R}_{Q^c}(\downarrow)$ | $\Phi_1$ | $\Phi_{N/2}$ | $\Phi_N$ |
| CL | SQLCODER2 | 100 | 48.9 / 100 | -48.9 | -29.5K | -59.1K | 100 | 36.8 / 100 | -36.8 | -31.3K | -62.7K | 100 | 41.0 / 100 | -41.0 | -39.4K | -78.8K |
| | +INF† | 28.2 | 39.3 / 12.8 | 40.2 | -4.7K | -9.4K | 71.9 | 35.5 / 28.8 | 31.6 | -12.4K | -24.8K | 6.1 | 35.3 / 6.8 | 44.1 | -2.5K | -5.0K |
| | +ERR† | 61.7 | 18.8 / 10.3 | 58.9 | -4.3K | -8.6K | 66.7 | 16.7 / 26.0 | 46.2 | -8.4K | -16.9K | 66.4 | 17.0 / 25.2 | 46.7 | -10.1K | -20.3K |
| | +INF†+ERR† | 18.1 | 8.3 / 1.0 | 56.5 | -442.2 | -942.2 | 49.9 | 16.2 / 7.9 | 59.0 | -3.6K | -7.2K | 4.5 | 12.0 / 2.7 | 49.0 | -849.4 | -1.7K |
| | GPT-4O | 100 | 12.3 / 92.2 | -4.5 | -20.7K | -41.5K | 100 | 5.0 / 97.8 | -2.7 | -23.5K | -47.1K | 100 | 10.7 / 94.5 | -5.2 | -29.4K | -58.8K |
| | +INF‡ | 78.8 | 13.7 / 11.3 | 67.3 | -4.3K | -8.7K | 52.3 | 6.2 / 9.4 | 63.5 | -2.8K | -5.7K | 62.6 | 9.4 / 5.9 | 69.5 | -3.2K | -6.5K |
| | +ERR‡ | 16.9 | 1.5 / 12.1 | 46.1 | -2.4K | -4.8K | 43.4 | 3.5 / 15.5 | 54.6 | -3.8K | -7.7K | 8.9 | 6.0 / 8.9 | 45.0 | -2.6K | -5.3K |
| | +INF‡+ERR‡ | 15.6 | 0.0 / 2.8 | 55.0 | -493.6 | -1.0K | 24.8 | 4.4 / 3.7 | 57.6 | -1.0K | -2.1K | 5.9 | 3.0 / 0.7 | 52.1 | -197.5 | -447.5 |
| UE | T5-3B | 100 | 53.9 / 100 | -53.9 | -30.5K | -61.1K | 100 | 36.2 / 100 | -36.1 | -31.1K | -62.4K | 100 | 42.2 / 100 | -42.2 | -39.7K | -79.5K |
| | +MAXENT | 14.6 | 0.0 / 2.0 | 55.3 | -343.7 | -743.7 | 6.8 | 3.2 / 2.0 | 51.1 | -447.2 | -947.8 | 11.1 | 0.0 / 0.5 | 55.0 | -94.7 | -244.7 |
| | +MAXPROB | 21.7 | 0.0 / 2.3 | 58.6 | -390.3 | -840.3 | 25.5 | 8.5 / 5.9 | 54.6 | -1.8K | -3.6K | 13.4 | 0.0 / 0.7 | 56.0 | -143.6 | -343.6 |
| | +FEATMD | 1.3 | 0.0 / 0.0 | 50.6 | 50.6 | 50.6 | 17.2 | 3.8 / 0.0 | 57.9 | -91.6 | -241.8 | 4.7 | 7.7 / 0.0 | 52.0 | -47.9 | -147.9 |
| | +FEATRMD | 0.8 | 0.0 / 0.0 | 50.4 | 50.4 | 50.4 | 16.1 | 0.0 / 0.0 | 58.0 | 58.0 | 58.0 | 1.4 | 12.5 / 0.0 | 50.5 | 0.6 | -49.4 |
| | GPT-4O | 100 | 12.3 / 92.2 | -4.5 | -20.7K | -41.5K | 100 | 5.0 / 97.8 | -2.7 | -23.5K | -47.1K | 100 | 10.7 / 94.5 | -5.2 | -29.4K | -58.8K |
| | +VERBAL | 68.5 | 8.8 / 15.1 | 63.1 | -4.1K | -8.3K | 84.5 | 3.1 / 30.3 | 59.3 | -7.5K | -15.0K | 78.7 | 6.8 / 23.1 | 60.9 | -7.9K | -15.8K |
| | +VOTE-SQL | 20.7 | 6.1 / 2.5 | 56.5 | -691.6 | -1.4K | 39.9 | 1.1 / 8.3 | 61.2 | -1.9K | -3.9K | 40.8 | 1.8 / 3.4 | 66.3 | -1.1K | -2.2K |
| | +VOTE-ANS | 82.1 | 4.9 / 18.6 | 68.4 | -4.4K | -8.9K | 51.0 | 5.1 / 17.5 | 55.4 | -4.5K | -9.1K | 82.6 | 4.3 / 22.9 | 64.8 | -7.3K | -14.7K |

scores in $\Phi_1$, but as the penalty increases, FT models using probability- or distance-based methods perform better in the higher-penalty RS. Only a few models achieve positive scores in $\Phi_N$, particularly the distance-based methods, indicating they answer only when confident in the correctness. However, this comes at the cost of extremely low $\mathcal{C}_Q$ (<2% for ATIS and <17% for Advising). For infeasible questions, `ambiguous` questions are the most challenging type to avoid, being the main cause of reduction in the RS, followed by `missing-schema` and `non-sql` (Table 12). For feasible questions, unfamiliar questions or `hard` questions are much more challenging to gain the RS(further shown in Appendix B.1).

## 6.2 FEASIBLE DATA, $D_Q$

Table 5 presents the performance of the models considering only $Q$ as the model inputs.

**Coverage-Risk** While the CL-based FT model (SQLCODER2+INF†+ERR†) demonstrates competitive performance in reducing errors on familiar, feasible questions (the first score in $\mathcal{R}_Q$), it does not perform as well as its ICL counterpart on unfamiliar questions (the second score in $\mathcal{R}_Q$), indicating limited generalization capability for a 7B model. Notably, the CL-based ICL model (GPT-4O+INF‡+ERR‡) and UE-based FT models achieve near-zero $\mathcal{R}_Q$, at the cost of low $\mathcal{C}_Q$ (averaging <16% and <12%, respectively). Methods like +VERBAL and +VOTE-ANS achieve higher $\mathcal{C}_Q$ (near 80%) while maintaining low $\mathcal{R}_Q$ (near 7%). Further details on $\mathcal{R}_Q$ for different query difficulties are in Appendix B.2.

**Reliability** The CL-based ICL model (GPT-4O+INF‡+ERR‡) and UE-based FT models often achieve near-zero $\mathcal{R}_Q$, resulting in positive scores in the RS across all penalty settings, meeting all user safety requirements. (Section 3). However, these models still fall short of achieving high text-to-SQL reliability (with a maximum of only about 22%), even when assuming all inputs are feasible, due to low $\mathcal{C}_Q$.

## 6.3 DISCUSSION

In this section, we address two key questions regarding the impact of abstention mechanisms on text-to-SQL and model selection using the RS.

**Q1: What is the impact of incorporating abstention mechanisms in the text-to-SQL task?**
Incorporating abstention mechanisms significantly changes how text-to-SQL models handle both

Table 5: Model comparison for only feasible data. For all metrics, higher values are better, except for $\mathcal{R}_Q$. The scores in the shaded cells represent SQL generation performance without abstention. For each $\mathcal{C}_Q$ and $\mathcal{R}_Q$, we report two separate scores: each for familiar and unfamiliar feasible questions (Section 4.4). All metrics are in % and numbers in thousands are abbreviated as K for readability.

| | | ATIS | | | | | Advising | | | | | EHRSQL | | | | |
|---|---|---|---|---|---|---|---|---|---|---|---|---|---|---|---|---|
| | | $\mathcal{C}_Q$ | $\mathcal{R}_Q(\downarrow)$ | $\Phi_1$ | $\Phi_{N/2}$ | $\Phi_N$ | $\mathcal{C}_Q$ | $\mathcal{R}_Q(\downarrow)$ | $\Phi_1$ | $\Phi_{N/2}$ | $\Phi_N$ | $\mathcal{C}_Q$ | $\mathcal{R}_Q(\downarrow)$ | $\Phi_1$ | $\Phi_{N/2}$ | $\Phi_N$ |
| | SQLCODER2 | 100 / 100 | 12.3 / 87.6 | 2.3 | -9.6K | -19.3K | 100 / 100 | 6.5 / 67.2 | 26.4 | -8.4K | -16.8K | 100 / 100 | 11.5 / 70.4 | 18.1 | -11.4K | -22.8K |
| | +INF† | 32.4 / 23.8 | 7.6 / 84.8 | 6.0 | -2.2K | -4.4K | 70.9 / 72.9 | 4.9 / 65.3 | 20.9 | -5.8K | -11.7K | 6.1 / 6.1 | 11.8 / 58.8 | 1.8 | -595.0 | -1.2K |
| | +ERR† | 91.2 / 30.6 | 4.8 / 62.7 | 38.5 | -2.2K | -4.5K | 92.6 / 40.6 | 1.4 / 51.6 | 44.4 | -2.5K | -5.0K | 88.2 / 44.6 | 2.4 / 45.6 | 43.8 | -3.1K | -6.2K |
| | +INF†+ERR† | 29.4 / 6.2 | 0.0 / 50.0 | 15.1 | -282.6 | -583.4 | 67.0 / 32.8 | 1.3 / 46.7 | 33.8 | -1.8K | -3.7K | 5.4 / 3.6 | 0.0 / 30.0 | 3.4 | -145.8 | -296.1 |
| CL | GPT-4O | 100 / 100 | 2.0 / 23.3 | 75.3 | -2.4K | -4.8K | 100 / 100 | 0.9 / 9.2 | 90.0 | -1.1K | -2.2K | 100 / 100 | 0.7 / 20.7 | 78.5 | -2.9K | -5.9K |
| | +INF‡ | 79.9 / 77.7 | 1.8 / 26.7 | 57.2 | -2.1K | -4.2K | 49.6 / 55.0 | 1.8 / 10.3 | 45.8 | -699.3 | -1.5K | 63.8 / 61.4 | 0.6 / 18.6 | 50.8 | -1.6K | -3.2K |
| | +ERR‡ | 20.1 / 13.5 | 0.0 / 3.8 | 16.4 | -33.2 | -83.4 | 44.3 / 42.4 | 0.0 / 7.2 | 40.3 | -307.4 | -658.2 | 12.5 / 5.4 | 0.0 / 20.0 | 7.9 | -141.3 | -291.6 |
| | +INF‡+ERR‡ | 19.1 / 11.9 | 0.0 / 0.0 | 15.6 | 15.6 | 15.6 | 24.8 / 24.9 | 1.8 / 7.0 | 22.7 | -225.7 | -476.3 | 8.2 / 3.6 | 0.0 / 10.0 | 5.5 | -44.2 | -94.3 |
| | T5-3B | 100 / 100 | 19.1 / 90.7 | -7.8 | -10.6K | -21.4K | 100 / 100 | 6.1 / 41.0 | 52.9 | -5.3K | -10.7K | 100 / 100 | 12.2 / 72.1 | 15.6 | -11.7K | -23.5K |
| | +MAXENT | 28.4 / 0.0 | 0.0 / 0.0 | 14.6 | 14.6 | 14.6 | 13.0 / 0.4 | 0.0 / 100 | 6.3 | -43.4 | -93.5 | 21.9 / 0.4 | 0.0 / 0.0 | 11.1 | 11.1 | 11.1 |
| | +MAXPROB | 42.2 / 0.0 | 0.0 / 0.0 | 21.7 | 21.7 | 21.7 | 46.1 / 4.8 | 0.0 / 54.5 | 22.9 | -275.2 | -575.8 | 26.5 / 0.4 | 0.0 / 0.0 | 13.4 | 13.4 | 13.4 |
| | +FEATMD | 2.5 / 0.0 | 0.0 / 0.0 | 1.3 | 1.3 | 1.3 | 33.0 / 1.3 | 1.3 / 0.0 | 16.8 | -32.9 | -83.0 | 9.3 / 0.0 | 7.7 / 0.0 | 3.9 | -95.5 | -195.7 |
| UE | +FEATRMD | 1.5 / 0.0 | 0.0 / 0.0 | 0.8 | 0.8 | 0.8 | 32.2 / 0.0 | 0.0 / 0.0 | 15.3 | -84.1 | -184.3 | 2.9 / 0.0 | 12.5 / 0.0 | 1.1 | -48.7 | -98.7 |
| | GPT-4O | 100 / 100 | 2.0 / 23.3 | 75.3 | -2.4K | -4.8K | 100 / 100 | 1.7 / 7.4 | 90.8 | -952.3 | -2.0K | 100 / 100 | 0.7 / 20.7 | 78.5 | -2.9K | -5.9K |
| | +VERBAL | 79.4 / 57.0 | 6.2 / 12.7 | 56.4 | -1.1K | -2.3K | 95.7 / 73.4 | 2.3 / 4.8 | 78.9 | -566.9 | -1.2K | 86.0 / 71.4 | 1.2 / 13.5 | 68.0 | -1.4K | -2.9K |
| | +VOTE-SQL | 21.6 / 19.7 | 0.0 / 13.2 | 18.1 | -230.0 | -480.6 | 54.3 / 25.3 | 0.0 / 1.7 | 39.4 | -10.2 | -60.3 | 57.7 / 23.9 | 0.0 / 6.0 | 39.4 | -159.6 | -359.9 |
| | +VOTE-ANS | 90.7 / 73.1 | 0.5 / 10.6 | 74.1 | -719.9 | -1.5K | 38.3 / 36.7 | 1.1 / 9.5 | 33.6 | -413.5 | -864.5 | 97.1 / 68.2 | 0.4 / 9.9 | 75.5 | -919.1 | -1.9K |

feasible and infeasible questions. Traditional approaches, such as applying infeasible question detectors before SQL generation, can effectively filter out infeasible questions but often exclude too many feasible questions during the SQL generation process, as shown in $\mathcal{C}_Q$ for models with +INF†‡ in Table 5. In contrast, integrating mechanisms like SQL error detection or uncertainty estimation during or after SQL generation allows more control over which questions stay until the end.

**Q2: What is the advantage of using the RS?** While coverage and risk provide an intuitive view of a model's performance in terms of the ratio of correctness, they alone may not be adequate measures of text-to-SQL reliability. For example, in scenarios where occasional mistakes are acceptable, models with higher coverage and some level of risk may be preferred. In this context, users can apply a lower penalty, such as $c = 1$, and select models with higher scores in $\Phi_1$ (GPT-4O+VOTE-ANS). In contrast, high-stakes scenarios like healthcare demand much stricter control over incorrect decisions. In such cases, models that show extremely low risk are prioritized, as even a single incorrect decision could lead to dire consequences. By applying a stricter penalty, such as $c = N$, the RS allows users to identify models that are more conservative but accurate in providing answers (GPT-4O+INF‡+ERR‡ or GPT-4O+VOTE-SQL). As a result, the coverage and risk required for a model to be helpful vary according to the user's safety requirements, but the RS provides users with a consistent measure that is positive if the model is helpful.

# 7 CONCLUSION

In this work, we introduce TrustSQL, a benchmark designed to evaluate the reliability of text-to-SQL models, prioritizing models that demonstrate helpfulness while minimizing harmfulness. To quantify this, we also propose a new metric, the Reliability Score (RS), which assigns positive scores for correct decisions to represent helpfulness and penalizes incorrect ones according to a user-defined safety penalty that accounts for harmfulness. TrustSQL is constructed by re-annotating three datasets—ATIS, Advising, and EHRSQL—and augmenting them with critical types of infeasible questions that may arise during deployment, enabling a comprehensive evaluation across diverse user inputs. We evaluate text-to-SQL models integrated with abstention mechanisms using both classifier-based and uncertainty estimation methods. Our experimental results show that only a few models achieve positive scores in the RS across all penalty settings, underscoring the need to not only improve SQL generation but also ensure a certain degree of reliability. We hope TrustSQL can shed light on developing more reliable text-to-SQL models, thus accelerating their adoption across vast industry areas in need.

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
