APPENDIX

# A  BENCHMARK DETAILS

## A.1  FEASIBLE QUESTIONS

Tables 6 and 7 show data statistics based on query difficulty and sample questions for feasible questions. For assigning query difficulty, we adopted the SQL hardness criteria from DIN-SQL (Pourreza & Rafiei, 2023) across all datasets, classifying the samples into three levels: `easy`, `medium`, and `hard`. The `easy` category includes single-table queries that do not require joins or nesting. The `medium` category includes queries that involve joins (including implicit joins) but excludes nesting. The `hard` category covers queries that may contain joins, sub-queries, and set operations.

Table 6: Data statistics based on query difficulty. *EHRSQL does not contain any medium-difficulty samples. We use the MIMIC-IV portion of the EHRSQL dataset for our work.

|  | ATIS | | | Advising | | | EHRSQL* | |
| --- | --- | --- | --- | --- | --- | --- | --- | --- |
|  | easy | medium | hard | easy | medium | hard | easy | hard |
| Train | 83 | 400 | 140 | 83 | 508 | 107 | 308 | 2325 |
| Valid | 37 | 259 | 111 | 67 | 309 | 75 | 79 | 480 |
| Test | 39 | 256 | 102 | 78 | 312 | 69 | 74 | 485 |
| All | 159 | 915 | 353 | 228 | 1129 | 251 | 461 | 3290 |

Table 7: Examples of feasible questions. Questions and their corresponding SQL queries in domain-specific datasets tend to be long, making it challenging for the model to handle such lengthy queries.

| Dataset | Difficulty | Question | SQL Query |
| --- | --- | --- | --- |
| ATIS | easy | Can you tell me the location of LGA? | SELECT DISTINCT AIRPORTalias0.AIRPORT_LOCATION FROM AIRPORT AS AIRPORTalias0 WHERE AIRPORTalias0.AIRPORT_CODE = "LGA" |
| | medium | Can you provide a list of flights that arrive at DAL? | SELECT DISTINCT FLIGHTalias0.FLIGHT_ID FROM AIRPORT AS AIRPORTalias0 , FLIGHT AS FLIGHTalias0 WHERE AIRPORTalias0.AIRPORT_CODE = "DAL" AND FLIGHTalias0.TO_AIRPORT = AIRPORTalias0.AIRPORT_CODE |
| | hard | What would be the cheapest flight from ATLANTA to DENVER on 10 / 12 / 1991? | SELECT DISTINCT SELECT DISTINCT FLIGHTalias0.FLIGHT_ID FROM AIRPORT_SERVICE AS AIRPORT_SERVICEalias0 , ... [38 lines omitted involving query nesting] ... FLIGHTalias0.FLIGHT_ID = FLIGHT_FAREalias0.FLIGHT_ID |
| Advising | easy | Give me the course number of the Investigations class. | SELECT DISTINCT COURSEalias0.NUMBER FROM COURSE AS COURSEalias0 WHERE COURSEalias0.NAME LIKE "%Investigations%" |
| | medium | Please provide me with the PreMajor courses that were available in Fall 2015. | SELECT DISTINCT COURSEalias0.DEPARTMENT , COURSEalias0.NUMBER FROM COURSE AS COURSEalias0 INNER JOIN COURSE_OFFERING AS COURSE_OFFERINGalias0 ... [9 lines omitted involving joins] ... = 2015 |
| | hard | Who, most recently, taught SCAND 104 before 2016? | SELECT DISTINCT INSTRUCTORalias0.NAME FROM COURSE AS COURSEalias0 INNER JOIN COURSE_OFFERING AS COURSE_OFFERINGalias0 ON COURSEalias0.COURSE_ID = ... [18 lines omitted involving query nesting] ... COURSEalias0.NUMBER = 104 |
| EHRSQL | easy | What are the ways to consume metformin (glucophage)? | SELECT DISTINCT prescriptions.route FROM prescriptions WHERE prescriptions.drug = 'metformin (glucophage)' |
| | hard | What was the change in arterial blood pressure systolic in patient 10037975 second measured on the first ICU visit compared to the first value measured on the first ICU visit? | SELECT ( SELECT chartevents.valuenum FROM chartevents WHERE chartevents.stay_id IN ( SELECT icustays.stay_id FROM icustays WHERE icustays.hadm_id IN ( SELECT admissions.hadm_id ... [15 lines omitted involving query nesting] ... ASC LIMIT 1 ) |

**SQL Assumptions**  Domain-specific datasets are commonly handled in a single-table setting (as in our work), which frequently relies on dataset-specific question-to-SQL assumptions (Suhr et al., 2020). During preprocessing and re-annotating the questions, we document these SQL mapping assumptions for each dataset in a separate file. This allows models using in-context learning (which typically requires a long context length) to reference the knowledge. Fine-tuned models can access this knowledge during training. Assumptions that are not explicitly stated or implicitly present in the training question-SQL pairs should not be used without the user's consent, as this could lead to potential harm. Below are the SQL assumptions for the three datasets.

### A.1.1 ATIS ASSUMPTIONS

- Use SQLite for SQL query generation.
- Unless otherwise specified, the values in the question are case-sensitive and are identical to the values stored in the database.
- When retrieving the result, use DISTINCT for the selected columns.
- For retrieving the top N results, exclude counts and return relevant items only.
- When asked to retrieve "flight," return "FLIGHT.FLIGHT_ID."
- When asked to retrieve "fares," return "FARE.FARE_ID."
- Treat phrases like "is it possible," "can you confirm whether," or "verify" as true/false questions.
- WEIGHT and PAY_LOAD are measured in pounds; CAPACITY refers to the number of passengers (seats); WING_SPAN and LENGTH are in feet; CRUISING_SPEED is in miles per hour; RANGE_MILES refers to miles.

### A.1.2 ADVISING ASSUMPTIONS

- Use SQLite for SQL query generation.
- - Unless otherwise specified, the values in the question are case-sensitive and are identical to the values stored in the database.
- When retrieving the result, use DISTINCT for the selected columns.
- When retrieving the top N results, return the relevant rows without including aggregate counts.
- Use "DISTINCT COURSE.DEPARTMENT, COURSE.NUMBER" when retrieving courses.
- Use "DISTINCT SEMESTER.SEMESTER, SEMESTER.YEAR" when retrieving information about semesters.
- Interpret questions involving "is it possible", "can you confirm whether", or "verify" as true/false queries.
- When asked about instructors, return "DISTINCT INSTRUCTOR.NAME".
- Differentiate between "course offering" (COURSE_OFFERING) and "course" (COURSE) when responding to queries.
- Use "AREA.AREA" for broader course categories such as theory, software, or intelligent systems; use "COURSE.NAME" for specific course names.
- Use a LIKE operator with "%" wildcards for queries involving "PROGRAM.NAME", "AREA.AREA", or "COURSE.DESCRIPTION".
- Map "Upper-level CS" to "ULCS" in the database.
- For columns containing "HAS_*", use "Y" for true conditions and "N" for false conditions.

### A.1.3 EHRSQL ASSUMPTIONS

- Use SQLite for SQL query generation.
- Use DENSE_RANK() for ranking results.
- For the top N results, return only the relevant items, excluding their counts.
- Use DISTINCT in queries asking about the cost of some events, drug routes, or counting patients or hospital/ICU visits.
- When calculating the total cost, compute the sum of patient's expenses for diagnoses, procedures, lab events, and prescriptions at a hospital admission level.
- The time of diagnosis occurs at the time of hospital admission, and procedures occur at the time of hospital discharge.
- Treat questions starts with "is it possible," "can you confirm," or "verify" as true/false questions.
- Calculate a patient's age once per hospital admission. The age remains constant even if the hospital stay exceeds one year.

- When referring to events related to inputs, the values must be found in the inputevents table; similarly, for output events, the values must be found in the outputevents table.
- Vital-related events are stored in chartevents, not in inputevents or outputevents.
- Values are distinct across tables. For example, "propofol" in the inputevents table does not appear in the prescription table.
- **When calculating the N-year survival rate**:
  - A patient is considered deceased if a death record exists between their first diagnosis and N years later.
  - If no death record exists within N years, or if death occurs afterward, the patient is considered survived.
  - Express the result as a percentage or proportion (values between 0 and 1).
- **Abnormality is defined only for the following values with these ranges**:
  - Temperature Celsius: (35.5, 38.1)
  - O2 saturation pulseoxymetry: (95.0, 100.0)
  - Heart Rate: (60.0, 100.0)
  - Respiratory Rate: (12.0, 18.0)
  - Arterial Blood Pressure systolic: (90.0, 120.0)
  - Arterial Blood Pressure diastolic: (60.0, 90.0)
  - Arterial Blood Pressure mean: (60.0, 110.0)
- **Time References**:
  - Interpret "now" as 2100-12-31 23:59:00, "today" as 2100-12-31, "this month" as December of 2100, and "this year" as 2100.
  - "3 months" refers to 365/4 days, and "6 months" refers to 365/2 days when calculating a duration.
  - "This month/02" and "last month/02" refer to the 2nd day of the current and previous month, respectively.
  - "11/this year" refers to November of this year; "11/2100 year" refers to November 2100; "12/31/this year" refers to December 31st of this year.
  - Dates like "12/06/2100" are in MM/DD/YYYY format.
  - When a question involves the time of diagnosis in relation to other types of events, use the first diagnosis time for each patient.
  - A "current hospital visit/encounter" refers to records where the hospital discharge time does not exist, indicating a current patient; a "last hospital visit" is only if the hospital discharge time is present; the first hospital visit is simply the first order of the visit.
- **Entity Mapping Assumptions**:
  - Use "temperature celsius" for body temperature.
  - Use "o2 saturation pulseoxymetry" for SaO2.
  - Use "heart rate" for heart rate.
  - Use "respiratory rate" for respiration rate.
  - Use "arterial blood pressure systolic" for systolic blood pressure.
  - Use "arterial blood pressure diastolic" for diastolic blood pressure.
  - Use "arterial blood pressure mean" for mean blood pressure.
  - Use "daily weight" for weight.
  - Use "height (cm)" for height.
  - Use "m" for male and "f" for female.
  - Except for the above, use the value as it is.

## A.2 Re-annotation Details

Our preliminary data analysis revealed that ATIS and Advising require significant preprocessing due to issues such as query duplication, low-quality natural language questions, null results (SQL queries returning empty results), and inconsistent question-to-SQL assumptions across samples (*e.g.*, SQL annotations with varying definitions of the current time). To address this we follow four data re-annotation steps below:

### A.2.1 Step 1: Template Review and Modification

**Template Sorting**  Domain-specific datasets contain unique SQL query structures that reflect user intents. For example, "I need a flight from `city_name1` to `city_name0`" and "Give me a list of flights going from `city_name1` to `city_name0`" are two different paraphrases for the same SQL structure. These datasets also include information about the entities or values used in each sample; for example, 'BOSTON' is a value for the 'city_name0' placeholder, which is sampled from a specific column. We refer to questions with placeholders as question templates. To validate the data, we first sort it by unique SQL structures.

**SQL Query Validation**  In cases of query duplication (such as in ATIS and Advising), where questions conveying the same intent are matched with differently labeled SQL structures (Finegan-Dollak et al., 2018), we merge these duplicates. Query duplication can be problematic for model evaluation, as it may lead to misleading results, particularly when assessing model performance on 'familiar' vs. 'unfamiliar' questions (Section A). To address this, we first identify samples that share identical placeholders (*e.g.*, "I need a flight from `city_name1` to `city_name0`" and "Which airline provides service in `city_name0` and `city_name1`" both use 'city_name0' and 'city_name1' as placeholders) and then assess their semantic equivalence. Once samples with identical semantics are found, we merge them by selecting the shortest SQL query to represent the group. In this step, we also verify whether the SQL queries accurately reflect the intended meanings of the corresponding questions.

### A.2.2 Step 2: Paraphrase Generation

Because ATIS, Advising, and EHRSQL are pre-GPT era datasets with all samples being human-curated, many of them are of low quality and sound unnatural, with limited diversity in the questions. To address this, we use GPT-4o to paraphrase at least 20 questions for each question template, ensuring that these paraphrases maintain the placeholders and align with the intended meanings. During this process, we make sure that each generated question does not assume any specific knowledge unique to individual questions. For example, in Advising, one question might assume 'I' refers to student 1, while another assumes 'I' refers to student 2. We eliminate such cases. Only the common assumptions that are applied extensively in SQL annotations are retained and documented.

### A.2.3 Step 3: Pair Construction

We merge question templates and paraphrases to construct a complete text-to-SQL dataset. During merging, we also insert sampled values (actual data stored in the database) into the placeholders in question paraphrases and SQL structures (now referred to SQL queries). Then, we execute these queries to see if we return null results from the database. Null results refer to cases where the SQL query returns 'Null' or '[]' upon execution, which may cause false positives in text-to-SQL model evaluation. If a null result is returned, we sample new values again from the database.

### A.2.4 Step 4: Review and Quality Control

All annotators engage in real-time discussion to resolve any disagreements in reviewing the created question-SQL pairs and SQL assumption text. Frequent disagreements involved determining whether the questions and the SQL assumptions included all necessary information to generate the annotated SQL queries. Discussions continue until a consensus is reached.

## A.3 INFEASIBLE QUESTIONS

There are three types of infeasible questions in TrustSQL: `missing-schema`, `ambiguous`, and `non-sql`. Our design choice is to make these questions similar to feasible questions in surface form while keeping it infeasible. So, we first define candidate keywords that assures infeasibility and ask annotators to incorporate that into existing feasible questions.

### A.3.1 MISSING-SCHEMA

To create questions that refer to non-existent columns, we first generate a list of 'hypothetical' columns that are similar to the database (which may not always be realistic). We create questions that refer to these hypothetical columns for each database by assuming they exist. For example: "Give me all aircraft types whose TURBOPROP engines are from HONEYWELL." Note that the Aircraft table in ATIS contains columns like 'manufacturer' and 'propulsion,' but no column named 'engine_manufacturer.' Below are the hypothetical columns we create for each database:

---

**Hypothetical Columns in ATIS**

**AIRCRAFT Table**:
COST, ENGINE_MANUFACTURER, FUEL_CAPACITY, MANUFACTURE_DATE, MAX_ALTITUDE, MAX_SPEED, PRODUCTION_STATUS

**AIRLINE Table**:
ANNUAL_PASSENGER_COUNT, EMPLOYEE_COUNT, FOUNDER, FOUNDING_YEAR, HEADQUARTERS, HUB_AIRPORT

**AIRPORT Table**:
ENTRANCE_DIRECTION, GATE_COUNT, NUM_LOUNGE, PARKING_CAPACITY, RUNWAY_LENGTH, TRANSIT_HOTEL

**FLIGHT Table**:
IN_FLIGHT_ENTERTAINMENT, WIFI_AVAILABLE

**FOOD_SERVICE Table**:
ALLERGENS, IS_VEGAN

**GROUND_SERVICE Table**:
OPERATIONAL_HOURS

**RESTRICTION Table**:
SUNDAY_STAY_REQUIREMENT

---

**Hypothetical Columns in Advising**

**COURSE Table**:
AP_IB_CREDIT_WAIVER, GRADING_METHOD, RATING

**COURSE_OFFERING Table**:
ATTENDANCE, DELIVERY_METHOD, GRADING_TYPE, HONORS_TRACK, LANGUAGE, TEXTBOOK_PUBLISHER, WAITLIST_COUNT

**INSTRUCTOR Table**:
AGE, IS_DEPARTMENT_CHAIR, OFFICE_ADDRESS, RATING, TOTAL_GPA

**STUDENT Table**:
ADVISOR, AGE, FINANCIAL_AID, HONOR_ROLL_STATUS, INTERNATIONAL_STUDENT

---

**STUDENT_RECORD Table**:
STUDY_ABROAD

**Hypothetical Columns in EHRSQL**

**admissions**:
age_at_marriage, arrival_time, attending_physician_id, department_id

**cost**:
billing_code, payment_status

**d_labitems**:
lab_priority

**inputevents**:
provider_id

**labevents**:
equipment, requester_id, turnaround_time

**microbiologyevents**:
requester_id, specimen_quality

**patients**:
address, blood_type, emergency_contact, insurance_duration,
is_veteran, name, next_of_kin, occupation, place_of_birth

**procedures_icd**:
anesthesia_duration, anesthesia_type, consent_date, physician_id,
procedure_duration

### A.3.2 AMBIGUOUS

For ambiguous questions, we have two sub-types: referential ambiguity and vagueness (Shah & Jinwala, 2015). This is different from existing works that focus solely on column ambiguity (*i.e.*, referring to multiple valid candidates for columns). However, since the definition of ambiguity involves being unable to answer without further clarification, this condition also holds. Questions with referential ambiguity include words where it is unclear who or what they refer to (*e.g.*, 'this,' 'she'). For example, "Has this patient been admitted to an emergency room?" (where 'this patient' is unclear in reference). Questions with vagueness contain words that are imprecise or lack clear boundaries (*e.g.*, 'early,' 'best'), such as in "Which upper-leval CS courses in Winter 2016 ended early?" (where 'early' is not precisely defined). We include this because a model should not rely on unstated commonsense to resolve vagueness, as it may conflict with the user's intent and cause potential harm. Any allowed assumptions should be explicitly stated (Appendix A.1.1, A.1.2, A.1.3) or be implicitly demonstrated in the question-SQL pairs within the training data.

Table 8 provides examples of infeasible questions due to ambiguity.

Table 8: Examples of ambiguous questions.

| Dataset | Question | Reason |
|---|---|---|
| ATIS | Show me the details of the flights that leave around 17:00. | Vagueness ("around" 17:00) |
| | Please give me the earliest time a flight takes off from BALTIMORE to BOSTON and the subsequent information. | Referential ambiguity ("subsequent information") |
| | What are the IDs of the fares of DL airline's flights from BOSTON to PHILADELPHIA that are economic? | Vagueness ("economic") |
| Advising | Who have been the instructors for CMPLXSYS 270 in recent years? | Vagueness ("recent" years) |
| | Which of those 200-level courses were 3-credit courses? | Referential ambiguity ("those" 200-level courses) |
| | What EECS courses are suitable for students whose total credits are less than 200? | Vagueness ("suitable") |
| EHRSQL | When was that first maximum pt for this patient in 12/this year? | Referential ambiguity ("this" patient) |
| | Can you show the five most commonly prescribed drugs for middle-aged patients? | Vagueness ("middle-aged") |
| | Can you find the patients who needed extra attention during their treatment? | Vaugeness ("extra attention") |

### A.3.3 NON-SQL

Non-SQL refers to questions that request operations or tasks outside SQL's standard scope. We use 12 types of non-SQL tasks as keywods across all datasets: 'Time Series Forecasting', 'Advanced Statistical Analysis', 'Feature Importance', 'Clustering', 'Causal Inference', 'Data Preprocessing', 'Outlier Detection', 'Web Search', 'Sentiment Analysis', 'Data-to-Text', 'Data Visualization', and 'Assistive Tasks'.

Table 9 shows examples of `non-sql` type questions.

Table 9: Examples of non-SQL questions.

| Dataset | Question | Reason |
|---------|----------|--------|
| ATIS | What is the earliest available flight from CHICAGO to NASHVILLE that includes breakfast service as found on current flight search engines? | Web Search |
| | Tell me the current status of my flight from BOSTON to NEW YORK. | Assistive Task |
| | How can you use a t-test to compare the average cost of Delta flights to other airlines for trips from Boston to Washington? | Advanced Statistical Analysis) |
| Advising | Can you create a scatter plot of course credits versus the clarity score for all EECS courses? | Data Visualization |
| | What is the textual summary of courses that started before 09:00 in Spring 2020? | Data-to-Text |
| | Can you encode the "AREA" column for all courses into word vectors? | Data Preprocessing |
| EHRSQL | What are the key predictors of successful weaning from mechanical ventilation in ICU patients? | Feature Importance |
| | Do you detect any signs of frustration in patient 10018354's records related to delayed procedures? | Sentiment Analysis |
| | Were there any seasonal patterns in drug prescriptions for patient admitted this year? | Time Series Analysis |

## A.4 COMPARISON OF KEYWORD-BASED AND TEMPLATE-BASED INFEASIBLE DATA CREATION

The keyword-based question creation method used in this work combines the strengths of template-based and template-free annotation methods. Annotators are provided with specific keywords and sample feasible questions. They modify these questions to make them infeasible by incorporating the keyword's intent. This approach introduces greater semantic diversity and creates infeasible questions that closely resemble real user questions, enhancing the dataset's realism and the model's ability to handle complex scenarios. Below are illustrative examples:

**Template-based method** Consider this infeasible question template: "When is the next earliest hospital visit of patient 0000?"—where no record exists for the next hospital appointment in the database. Possible paraphrases generated from this template can be the following:

- "What is the soonest upcoming hospital visit scheduled for patient 0000?"
- "When is the next scheduled appointment for patient 0000?"

While this method can effectively handle common unanswerable questions (missing-schema), the diversity of the question pool generated using this method may be limited.

**Keyword-based method** Suppose the keyword 'appointment' is provided, along with sample feasible questions. The task is to modify these questions to include the keyword, making them infeasible.

Sampled feasible questions:

- "Has patient 0000 gotten any medication this year?"
- "Provide the count of hospital visits for patient 0000."

Annotated infeasible questions (infeasible keyword is now inserted):

- "Has patient 0000 gotten any medication this year and do they have any upcoming appointments?"
- "Provide the count of hospital visits for patient 0000 including any scheduled upcoming appointments."

By incorporating the keyword into existing feasible questions to make them infeasible, this method ensures both semantic diversity and guarantees that the annotated questions are indeed infeasible. This approach allows us to create a wider range of infeasible questions that closely resemble real user questions.

# B DETAILED PERFORMANCE

## B.1 DETAILED TASK PERFORMANCE

Tables 10 and 11 present the RS of the models on familiar and unfamiliar questions in the feasible data, respectively. Additionally, Table 12 shows the RS of the models on infeasible questions.

As expected, the models generally perform better on familiar questions than on unfamiliar ones. Notably, the T5-3B model using the +MAXENT and +MAXPROB methods achieved non-negative RS even under the strictest safety requirements for familiar questions by abstaining from answering both unfamiliar feasible questions and infeasible ones. This suggests that less powerful models like T5-3B may be more reliable when they only answer questions they are confident about, thereby avoiding the risk of generating incorrect SQL queries. We also observed a correlation between the RS and query difficulty, indicating that models tend to have higher reliability scores on simpler queries. When detecting infeasible questions, most models found `ambiguous` questions to be the most challenging. This highlights the need for methods that enable models to detect subtle ambiguities to improve reliability.

| | | ATIS | | Advising | | EHRSQL | |
| --- | --- | --- | --- | --- | --- | --- | --- |
| | | $\Phi_1$ (easy / medium / hard) | $\Phi_N$ (easy / medium / hard) | $\Phi_1$ (easy / medium / hard) | $\Phi_N$ (easy / medium / hard) | $\Phi_1$ (easy / hard) | $\Phi_N$ (easy / hard) |
| CL | SQLCODER2 | 100 / 68.4 / 81.8 | 100 / -12.5K / -7.1K | 92.9 / 91.4 / 79.5 | -3.2K / -3.8K / -9.3K | 100 / 72.8 | 100 / -15.1K |
| | +INF[†] | 18.5 / 24.1 / 43.2 | 18.5 / -3.0K / 43.2 | 75.0 / 65.0 / 66.7 | -3.2K / -2.2K / 66.7 | 11.4 / 3.4 | 11.4 / -947.2 |
| | +ERR[†] | 100 / 78.2 / 84.1 | 100 / -4.1K / -3.5K | 89.3 / 89.6 / 87.2 | 89.3 / -1.6K / -2.3K | 100 / 80.9 | 100 / -2.8K |
| | +INF[†]+ERR[†] | 18.5 / 27.8 / 40.9 | 18.5 / 27.8 / 40.9 | 71.4 / 62.6 / 66.7 | 71.4 / -1.6K / 66.7 | 11.4 / 4.3 | 11.4 / 4.3 |
| | GPT-4O | 100 / 94.0 / 100 | 100 / -2.3K / 100 | 100 / 95.1 / 100 | 100 / -2.2K / 100 | 100 / 98.3 | 100 / -852.3 |
| | +INF[†] | 88.9 / 78.9 / 63.6 | 88.9 / -1.7K / 63.6 | 60.7 / 43.6 / 46.2 | 60.7 / -2.2K / 46.2 | 97.7 / 56.6 | 97.7 / -418.7 |
| | +ERR[†] | 63.0 / 16.5 / 4.5 | 63.0 / 16.5 / 4.5 | 67.9 / 45.4 / 17.9 | 67.9 / -516.6 / 17.9 | 13.6 / 12.3 | 13.6 / 12.3 |
| | +INF[‡]+ERR[‡] | 63.0 / 15.0 / 4.5 | 63.0 / 15.0 / 4.5 | 39.3 / 22.7 / 17.9 | 39.3 / -539.3 / 17.9 | 13.6 / 7.2 | 13.6 / 7.2 |
| UE | T5-3B | 100 / 66.9 / 22.7 | 100 / -13.1K / -30.6K | 92.9 / 93.9 / 79.5 | -3.2K / -2.7K / -9.3K | 100 / 71.1 | 100 / -16.1K |
| | +MAXENT | 81.5 / 19.5 / 22.7 | 81.5 / 19.5 / 22.7 | 32.1 / 12.3 / 2.6 | 32.1 / 12.3 / 2.6 | 47.7 / 17.0 | 47.7 / 17.0 |
| | +MAXPROB | 100 / 32.3 / 36.4 | 100 / 32.3 / 36.4 | 71.4 / 46.6 / 25.6 | 71.4 / 46.6 / 25.6 | 63.6 / 19.6 | 63.6 / 19.6 |
| | +FEATMD | 11.1 / 1.5 / 0.0 | 11.1 / 1.5 / 0.0 | 32.1 / 33.7 / 25.6 | 32.1 / -528.2 / 25.6 | 9.1 / 7.7 | 9.1 / -943.0 |
| | +FEATRMD | 0.0 / 0.0 / 6.8 | 0.0 / 0.0 / 6.8 | 25.0 / 33.1 / 23.1 | 25.0 / -1.1K / 23.1 | 2.3 / 2.1 | 2.3 / -473.2 |
| | GPT-4O | 100 / 94.0 / 100 | 100 / -2.3K / 100 | 100 / 95.1 / 100 | 100 / -2.2K / 100 | 100 / 98.3 | 100 / -852.3 |
| | +VERBAL | 70.4 / 70.7 / 65.9 | 70.4 / -5.9K / 65.9 | 89.3 / 93.3 / 84.6 | 89.3 / -2.7K / 84.6 | 88.6 / 83.0 | -2.5K / -867.7 |
| | +VOTE-SQL | 33.3 / 22.6 / 11.4 | 33.3 / 22.6 / 11.4 | 50.0 / 53.4 / 61.5 | 50.0 / 53.4 / 61.5 | 72.7 / 54.9 | 72.7 / 54.9 |
| | +VOTE-ANS | 96.3 / 88.0 / 90.9 | 96.3 / -508.3 / 90.9 | 39.3 / 42.3 / 15.4 | 39.3 / -519.6 / 15.4 | 100 / 95.7 | 100 / -379.6 |

Table 10: Model comparison for 'familiar' feasible questions by query difficulty. The scores in the shaded cells represent SQL generation performance without abstention. All metrics are in % and numbers in thousands are abbreviated as K for readability.

| | | ATIS | | Advising | | EHRSQL | |
|---|---|---|---|---|---|---|---|
| | | $\Phi_1$ (easy / medium / hard) | $\Phi_N$ (easy / medium / hard) | $\Phi_1$ (easy / medium / hard) | $\Phi_N$ (easy / medium / hard) | $\Phi_1$ (easy / hard) | $\Phi_N$ (easy / hard) |
| | SQLCoder2 | -100 / -80.5 / -58.6 | -79.4K / -71.6K / -63.0K | -32.0 / 23.5 / 33.3 | -60.5K / -35.0K / -30.5K | -60.0 / -38.4 | -89.4K / -77.3K |
| | +Inf† | 0.0 / -29.3 / 6.9 | 0.0 / -23.2K / -4.1K | -40.0 / 24.8 / 26.7 | -53.2K / -21.5K / -21.3K | -10.0 / 0.0 | -18.6K / -2.2K |
| | +Err† | -8.3 / -11.4 / 0.0 | -6.6K / -16.8K / -13.7K | -10.0 / 28.2 / 33.3 | -9.2K / -9.8K / -6.1K | -10.0 / 5.6 | -33.5K / -21.4K |
| CL | +Inf†+Err† | 0.0 / -4.9 / 10.3 | 0.0 / -3.9K / 10.3 | -10.0 / 26.2 / 23.3 | -9.2K / -6.1K / -6.1K | 0.0 / 1.6 | -7.4K / -445.2 |
| | GPT-4o | 100 / 49.6 / 51.7 | 100 / -19.9K / -19.1K | 88.0 / 85.2 / 86.7 | -5.4K / -6.7K / -6.0K | 86.7 / 55.2 | -7.4K / -25.0K |
| | +Inf† | 75.0 / 33.3 / 34.5 | 75.0 / -16.7K / -19.1K | 48.0 / 51.7 / 36.7 | -5.4K / -1.2K / -6.1K | 83.3 / 33.2 | 83.3 / -14.3K |
| | +Err† | 91.7 / 10.6 / 0.0 | 91.7 / -634.1 / 0.0 | 68.0 / 34.9 / 3.3 | -5.4K / -1.2K / 3.3 | 6.7 / 2.8 | -3.7K / -890.8 |
| | +Inf‡+Err‡ | 75.0 / 11.4 / 0.0 | 75.0 / 11.4 / 0.0 | 36.0 / 20.8 / 0.0 | -5.5K / -594.0 / 0.0 | 10.0 / 2.0 | 10.0 / -444.8 |
| | T5-3B | -100 / -80.5 / -79.3 | -79.4K / -71.6K / -71.2K | -16.0 / 31.5 / 40.0 | -53.1K / -31.3K / -27.4K | -26.7 / -46.4 | -70.8K / -81.8K |
| | +MaxEnt | 0.0 / 0.0 / 0.0 | 0.0 / 0.0 / 0.0 | 0.0 / 0.7 / 0.0 | 0.0 / 0.7 / 0.0 | 0.0 / 0.4 | 0.0 / 0.4 |
| | +MaxProb | 0.0 / 0.0 / 0.0 | 0.0 / 0.0 / 0.0 | -8.0 / 4.7 / 0.0 | -7.3K / 4.7 / 0.0 | 0.0 / 0.4 | 0.0 / 0.4 |
| | +FeatMD | 0.0 / 0.0 / 0.0 | 0.0 / 0.0 / 0.0 | 0.0 / 2.0 / 0.0 | 0.0 / 2.0 / 0.0 | 0.0 / 0.0 | 0.0 / 0.0 |
| UE | +FeatRMD | 0.0 / 0.0 / 0.0 | 0.0 / 0.0 / 0.0 | 0.0 / 0.0 / 0.0 | 0.0 / 0.0 / 0.0 | 0.0 / 0.0 | 0.0 / 0.0 |
| | GPT-4o | 100 / 49.6 / 51.7 | 100 / -19.9K / -19.1K | 88.0 / 85.2 / 86.7 | -5.4K / -6.7K / -6.0K | 86.7 / 55.2 | -7.4K / -25.0K |
| | +Verbal | 91.7 / 39.8 / 37.9 | 91.7 / -8.3K / -1.3K | 74.0 / 68.5 / 43.3 | -7.3K / -2.4K / 43.3 | 80.0 / 48.8 | -3.6K / -11.6K |
| | +Vote-SQL | 50.0 / 13.8 / 8.6 | 50.0 / -1.3K / -4.1K | 38.0 / 23.5 / 6.7 | -1.8K / 23.5 / 6.7 | 40.0 / 18.8 | 40.0 / -1.8K |
| | +Vote-ANS | 91.7 / 52.8 / 60.3 | 91.7 / -6.4K / -6.8K | 30.0 / 31.5 / 0.0 | -5.5K / -3.0K / 0.0 | 86.7 / 48.4 | 86.7 / -8.4K |

Table 11: Model comparison for 'unfamiliar' feasible questions by query difficulty. The scores in the shaded cells represent SQL generation performance without abstention. All metrics are in % and numbers in thousands are abbreviated as K for readability.

| | | ATIS | | Advising | | EHRSQL | |
|---|---|---|---|---|---|---|---|
| | | $\Phi_1$ (missing / ambig / non-sql) | $\Phi_N$ (missing / ambig / non-sql) | $\Phi_1$ (missing / ambig / non-sql) | $\Phi_N$ (missing / ambig / non-sql) | $\Phi_1$ (missing / ambig / non-sql) | $\Phi_N$ (missing / ambig / non-sql) |
| | SQLCoder2 | -100 / -100 / -100 | -79.4K / -79.4K / -79.4K | -100 / -100 / -100 | -91.7K / -91.7K / -91.7K | -100 / -100 / -100 | -111.8K / -111.8K / -111.8K |
| | +Inf† | 82.0 / 59.1 / 81.8 | -7.1K / -16.2K / -7.1K | 71.1 / 9.8 / 46.4 | -13.2K / -41.3K / -24.5K | 91.4 / 77.4 / 90.3 | -4.7K / -12.5K / -5.3K |
| | +Err† | 95.5 / 56.1 / 86.4 | -1.7K / -17.4K / -5.3K | 52.6 / 28.1 / 63.4 | -21.6K / -32.9K / -16.7K | 74.3 / 6.5 / 67.7 | -14.3K / -52.2K / -17.9K |
| CL | +Inf†+Err† | 98.5 / 95.5 / 100 | -497.7 / -1.7K / 100 | 90.8 / 71.2 / 90.8 | -4.1K / -13.1K / -4.1K | 97.9 / 86.0 / 100 | -1.1K / -7.7K / 100 |
| | GPT-4o | -72.9 / -97.0 / -83.3 | -68.6K / -78.2K / -72.8K | -92.1 / -98.7 / -96.1 | -88.1K / -91.1K / -89.9K | -91.4 / -97.8 / -77.4 | -107.0K / -110.6K / -99.2K |
| | +Inf‡ | 71.4 / 66.7 / 93.9 | -11.3K / -13.2K / -2.3K | 71.1 / 73.9 / 98.7 | -13.2K / -11.9K / -500.0 | 90.4 / 78.5 / 95.7 | -5.3K / -11.9K / -2.3K |
| | +Err‡ | 74.4 / 72.7 / 80.3 | -10.1K / -10.7K / -7.7K | 71.1 / 62.1 / 73.9 | -13.2K / -17.3K / -11.9K | 86.1 / 87.1 / 73.1 | -7.7K / -7.1K / -14.9K |
| | +Inf‡+Err‡ | 86.5 / 97.0 / 100 | -5.3K / -1.1K / 100 | 89.5 / 89.5 / 98.7 | -4.7K / -4.7K / -500.0 | 97.9 / 98.9 / 98.9 | -1.1K / -501.6 / -501.6 |
| | T5-3B | -100 / -100 / -100 | -79.4K / -79.4K / -79.4K | -100 / -100 / -100 | -91.7K / -91.7K / -91.7K | -100 / -100 / -100 | -111.8K / -111.8K / -111.8K |
| | +MaxEnt | 98.5 / 92.4 / 97.0 | -497.7 / -2.9K / -1.1K | 97.4 / 93.5 / 97.4 | -1.1K / -2.9K / -1.1K | 98.9 / 97.8 / 100 | -498.4 / -1.1K / 100 |
| | +MaxProb | 97.0 / 92.4 / 97.0 | -1.1K / -2.9K / -1.1K | 90.8 / 79.1 / 94.8 | -4.1K / -9.5K / -2.3K | 97.9 / 97.8 / 100 | -1.1K / -1.1K / 100 |
| | +FeatMD | 100 / 100 / 100 | 100 / 100 / 100 | 100 / 100 / 100 | 100 / 100 / 100 | 100 / 100 / 100 | 100 / 100 / 100 |
| UE | +FeatRMD | 100 / 100 / 100 | 100 / 100 / 100 | 100 / 100 / 100 | 100 / 100 / 100 | 100 / 100 / 100 | 100 / 100 / 100 |
| | GPT-4o | -72.9 / -97.0 / -83.3 | -68.6K / -78.2K / -72.8K | -92.1 / -98.7 / -96.1 | -88.1K / -91.1K / -89.9K | -91.4 / -97.8 / -77.4 | -107.0K / -110.6K / -99.2K |
| | +Verbal | 60.9 / 57.6 / 90.9 | -15.4K / -16.8K / -3.5K | 32.9 / 20.3 / 64.7 | -30.7K / -36.5K / -16.1K | 50.8 / 22.6 / 88.2 | -27.4K / -43.2K / -6.5K |
| | +Vote-SQL | 92.5 / 93.9 / 98.5 | -2.9K / -2.3K / -502.3 | 81.6 / 75.2 / 93.5 | -8.4K / -11.3K / -2.9K | 94.7 / 87.1 / 97.8 | -2.9K / -7.1K / -1.1K |
| | +Vote-ANS | 63.9 / 57.6 / 66.7 | -14.2K / -16.8K / -13.2K | 43.4 / 39.9 / 79.1 | -25.9K / -27.5K / -9.5K | 60.4 / 31.2 / 72.0 | -22.0K / -38.4K / -15.5K |

Table 12: Model comparison for infeasible data, categorized by the types on infeasibe questions. 'missing' and 'ambig' refer to `missing-schema` and `ambiguous`, respectively. All metrics are in % and numbers in thousands are abbreviated as K for readability.

## B.2 SQL GENERATION PERFORMANCE

| | ATIS | | Advising | | EHRSQL | |
|---|---|---|---|---|---|---|
| | familiar (all \| easy / medium / hard) | unfamiliar (all \| easy / medium / hard) | familiar (all \| easy / medium / hard) | unfamiliar (all \| easy / medium / hard) | familiar (all \| easy / hard) | unfamiliar (all \| easy / hard) |
| T5-3B | 80.9 \| 100 / 83.5 / 61.4 | 9.3 \| 0.0 / 9.8 / 10.3 | 93.9 \| 82.1 / 96.9 / 89.7 | 33.6 \| 38.0 / 35.6 / 16.7 | 87.8 \| 100 / 85.5 | 27.9 \| 36.7 / 26.8 |
| SQLCODER2 | 87.7 \| 100 / 84.2 / 90.9 | 12.4 \| 0.0 / 9.8 / 20.7 | 93.5 \| 89.3 / 95.1 / 89.7 | 32.8 \| 20.0 / 40.9 / 13.3 | 88.5 \| 100 / 86.4 | 29.6 \| 20.0 / 30.8 |
| GPT-4O | 98.0 \| 100 / 97.0 / 100 | 76.7 \| 100 / 74.8 / 75.9 | 99.1 \| 100 / 99.4 / 97.4 | 90.8 \| 92.0 / 90.6 / 90.0 | 99.3 \| 100 / 99.1 | 79.3 \| 93.3 / 77.6 |
| DIN-SQL (Pourreza & Rafiei, 2023) | 67.6 \| 88.9 / 65.4 / 61.4 | 45.1 \| 100 / 40.7 / 43.1 | 77.8 \| 96.4 / 74.8 / 76.9 | 74.7 \| 94.0 / 66.4 / 83.3 | 69.5 \| 93.2 / 65.1 | 66.1 \| 76.7 / 64.8 |
| MAC-SQL (Wang et al., 2023c) | 72.1 \| 88.9 / 71.4 / 63.6 | 65.3 \| 100 / 67.5 / 53.4 | 79.1 \| 82.1 / 85.9 / 48.7 | 76.4 \| 82.0 / 77.9 / 60.0 | 68.8 \| 88.6 / 65.1 | 65.7 \| 83.3 / 63.6 |

Table 13: Performance of SQL generators in execution accuracy. DIN-SQL and MAC-SQL also use GPT-4o.

Table 13 shows the execution accuracy of the baseline SQL generators used in the experiments, along with two state-of-the-art text-to-SQL models (*i.e.*, DIN-SQL and MAC-SQL). GPT-4O outperforms on unfamiliar questions, demonstrating better generalizability than T5-3B and SQL-CODER2. Since DIN-SQL and MAC-SQL are designed for cross-database settings (with extra effort put into adapting their prompts based on TrustSQL samples for these experiments), they still underperform compared to GPT-4O, which benefits from retrieving relevant question-to-SQL pairs from the training set for a given input question.

## C  MODEL IMPLEMENTATION

### C.1  FINE-TUNING

#### C.1.1  CLASSIFIER-BASED MODELS

**SQL Generator**  For SQLCODER2, we include both the question and the database schema, formatted in DIN-SQL. We fine-tune a SQLCoder 7B-2 model for 20 epochs with a learning rate of $2 \times 10^{-4}$, using two NVIDIA A100 GPUs.

**Infeasible Question Detector**  We fine-tune another SQLCoder 7B-2 model (+INF$^{\dagger}$) to classify input question types using the TriageSQL dataset (Zhang et al., 2020), which contains five types: 'small talk', 'ambiguous', 'lack of data', 'unanswerable by SQL', and 'answerable'. Since the raw data is imbalanced, we sample 10,000 instances from each category to create a balanced training set. The model is fine-tuned for 1 epoch using two NVIDIA A100 GPUs. We consider a question to be infeasible if the model predicts any type other than 'answerable'.

**SQL Error Detector**  We also utilize SQLCoder 7B-2 for detecting SQL errors (+ERR$^{\dagger}$). Specifically, we generate training data based on the sample-level accuracy of SQL generation by SQL-CODER2 on the validation set. For each sample, we provide the question, the generated SQL, and the database schema to the model, and fine-tune it to predict whether the generated SQL is valid. The model is trained for 10 epochs with a learning rate of $2 \times 10^{-4}$, using two NVIDIA A100 GPUs.

#### C.1.2  UNCERTAINTY-ESTIMATION-BASED MODELS

**SQL Generator**  For training T5-3B, we prepare the input data as questions followed by a serialized schema that lists all tables and columns for each database (Suhr et al., 2020). We fine-tune the model using BF16 precision and the Adam optimizer with a learning rate of $1 \times 10^{-4}$, continuing the training until the validation loss ceases to decrease. The training is conducted using NVIDIA RTX A6000 GPUs.

**Abstention Through Uncertainty Estimation**  We leverage T5-3B to implement four uncertainty estimation methods: +MAXENT, +MAXPROB, FEATMD, and FEATRMD. Specifically, +MAXENT uses the entropy of the model's output distribution as an uncertainty measure, while +MAXPROB relies on the maximum probability from the output distribution. For FEATMD and FEATRMD, which are feature-based methods, we use the encoder representations as features to compute the Mahalanobis distance and relative Mahalanobis distance, respectively. In our preliminary analysis, we found that methods utilizing features from the decoder and binary logits, as proposed in (Ren et al., 2022), did not perform as effectively as these baselines.

Unlike most uncertainty estimation works, which report performance based on the area under the ROC curve (AUROC), TrustSQL requires setting a specific threshold to decide whether to provide answers from the generated SQL or to abstain. We apply an automatic threshold selection method to all uncertainty-estimation-based models, using a penalty parameter $c$, as outlined below:

1. For each sample in the validation set, assign a score of $+1$ if the model's decision is correct (i.e., the generated SQL is accurate), and a score of $-c$ if the decision is incorrect.

2. Sort the samples in descending order based on their confidence scores (e.g., lower entropy corresponds to higher confidence). This assumes that samples with higher confidence are more likely to result in correct SQL generation.

3. Compute the cumulative sum of the scores. Select the threshold at which the cumulative score stops increasing. If multiple thresholds yield the same maximum cumulative score, choose the one corresponding to the higher confidence level. This process is illustrated in Figure 2.

4. Apply this threshold during inference on the test data.

| Cumulative score: | +1 +2 +3 | +2 +3 | +2 +3 +2 | +3 | +2 | +1 | +0 | -1 -2 |

Figure 2: Samples sorted in descending confidence scores, with 14 samples and $c = 1$ for illustrative purposes.

## C.2 IN-CONTEXT LEARNING

For in-context learning, we leverage GPT-4o[5] as the backbone LLM.

**SQL Generator**   We solve TrustSQL in a single-database setting using GPT-4o (GPT-4O), where we are allowed to use samples from the training data. As a result, we retrieve eight question-SQL pairs selected based on the maximal marginal relevance (MMR) criterion. MMR ensures a balance between relevance and diversity, helping us select question-SQL pairs that are both highly relevant to the input question and diverse enough to avoid redundancy.

> Task: Translate the following questions into corresponding SQL queries based on the database schema provided below.
>
> Database Schema:
> {database_schema}
>
> Foreign Keys:
> {foreign_keys}
>
> SQL Assumptions:
> {sql_assumption}
>
> Question: {question1}
> SQL: {sql1}
>
> ...
>
> Question: {question8}
> SQL: {sql8}
>
> Question: {question}
> SQL:

### C.2.1 CLASSIFIER-BASED MODELS

**Infeasible Question Detector**   We employ zero-shot chain-of-thought prompting, providing definitions for the three categories of infeasible questions, followed by the database schema, to generate either 'Feasible' or 'Infeasible' responses (+INF[‡]). The model follows a reasoning step to arrive at its decision.

> Task: Given a natural language question, classify it as either "feasible" or "infeasible" for generating a corresponding SQL query.
>
> Definitions:
> - Feasible: The question can be accurately translated into an SQL query using the information available in the database.
> - Infeasible: The question cannot be translated into an SQL query due to intrinsic limitations, such as

---

[5]gpt-4o-2024-08-06

a lack of context, incompatibility with the database schema, or the inherent limitations of SQL.

Consider the following cases when determining infeasibility:
- Missing Schema (missing-schema): The question refers to database elements (e.g., tables, columns) that are not present in the provided schema.
- Query Ambiguity (ambiguous): The question contains vague, ambiguous, or subjective terms, making it unclear how to map or filter the requested information into an SQL query.
- Non-SQL (non-sql): The question requires operations or tasks (e.g., machine learning, advanced statistical analysis, deep domain expertise) that cannot be performed using standard SQL capabilities.

Database Schema:
{database_schema}

Assumptions:
{sql_assumption}

=====

Now, determine if the following question is "feasible" or "infeasible" to generate a corresponding SQL query.

Response Format:

```
{
    "chain-of-thought-reasoning" : "Explain your reasoning here.",
    "answer" : "feasible or infeasible"
}
```
Question: {question}

**SQL Error Detector**    We adapt DIN-SQL's zero-shot self-correction prompt to generate 'Correct' or 'Incorrect' labels based on the input question and the SQL output from GPT-4O (+ERR[‡]). The prompt is shown below:

Task: Based on the question and predicted SQL, are you sure the SQL below is correct? If you consider the SQL is correct, answer me with 'correct'. If not, answer me with 'incorrect'. Only output your response without explanation.

Database Schema:
{database_schema}

Assumptions:
{sql_assumption}

Foreign_keys = {foreign_keys}

Primary_keys = {primary_keys}

Question: {question}
Predicted SQL: {predicted_sql}
Answer:

### C.2.2    UNCERTAINTY-ESTIMATION-BASED MODELS

For +VOTE-SQL and +VOTE-ANS, we use the same prompt as GPT-4O but sample five times to check the consistency of the output.

For +VERBAL, we also use the same prompt, but providing both eight examples of correct and incorrect question-SQL pairs. Incorrect pairs are created by randomly pairing questions with unrelated SQL queries. Each pair is then labeled as either 'True' or 'False,' depending on its validity.