# OpenReview forum: "TrustSQL: Benchmarking Text-to-SQL Reliability with Penalty-Based Scoring"
_ICLR.cc/2025/Conference — ICLR 2025 Conference Withdrawn Submission_

### Official Review · Reviewer_xD5x · 2024-10-22

**Soundness:** 1
**Presentation:** 1
**Contribution:** 1
**Rating:** 1
**Confidence:** 5

**Summary:**

This paper introduces TrustSQL, a new benchmark for evaluating text-to-SQL model reliability. Current benchmarks overlook real-world scenarios with unanswerable questions, and existing models miss abstention mechanisms. TrustSQL addresses these issues by including both answerable and unanswerable questions, introducing a new scoring system rewarding correct answers and abstentions while penalizing errors. To faciliate evaluation, authors re-annotated three datasets and added infeasible questions to formalize a comprehensive benchmark. Experiments and related analysis reveal potential issues of current text-to-SQL systems when facing such safety requirements.

**Strengths:**

1) This paper proposes an important research problem of infeasible questions in the text-to-SQL domain.
2) Authors introduce a novel evaluation metric to assess model reliability regarding the potential harm of infeasible questions.
3) The work incorporates these features by re-annotating three existing datasets.
4) Experiments demonstrate the impact of abstention mechanisms and the Reliability Score in evaluating text-to-SQL model reliability.

**Weaknesses:**

### Unclear Dataset Construction:
1) The paper re-annotates three existing datasets (ATIS, Advising, and EHRSQL), which are limited. The generalizability of the findings across other domains or more diverse database schemas is uncertain. In contrast, benchmarks like SPIDER and BIRD include test sets with broader domain coverage (e.g., > 10 domains), making the generalizability of the dataset in this paper more limited in comparison.

2) The creation of "infeasible questions" appears to rely on manual annotation, potentially introducing bias or inconsistency. However, the authors do not provide **detailed guidelines** on the generation process for these infeasible questions. Appendix D only offers a high-level overview of data re-annotation without elaborating on **specific types** of infeasible questions or **how these types were defined and re-annotated**. Discussing measures taken to ensure consistency across annotators and mitigate potential biases in the manual annotation process would be more helpful. Additionally, while the authors criticize related work (e.g., EHR-SQL [2]) for using a **template-based** method to generate infeasible questions, their approach similarly employs **templates** (as suggested in Lines 1362–1363). This raises the question of how their method fundamentally differs from EHR-SQL. Furthermore, **no comparison** of data distributions between the authors’ dataset and related work, such as [1] and [2], is provided. Both of these works offer clear, detailed taxonomies of ambiguous with distributions or unanswerable (infeasible) question types, which is missed in this paper. The single table provided (Table 1) and Section 4.2 are quite insufficient; more detailed charts or a comprehensive taxonomy are necessary, especially since this is a central problem the paper aims to address. Also authors just state which papers that they follow to define for each category without discussing motivations of each, and it also seems the contribution of this work is incremental and limited.

3) When constructing benchmarks, it is essential to clearly describe the process of crowdsourcing. The paper should detail how many annotators were involved, what their expertise levels were, how the recruitment process worked, and what methods were used for data evaluation. Were expert panels involved? What were the criteria for evaluating the quality of annotations, and how was the workflow structured? Additionally, without information on compensations or incentives for crowdsourcing, it is difficult to evaluate the overall quality and reliability of the benchmark. These crucial details are missing, making it harder for readers to trust the quality and fairness of the dataset.

4) The paper does not adequately differentiate its contributions from similar works. Both [1] and [2] have already addressed infeasible questions (often referred to as ambiguous or unanswerable questions). The distinction in this work between their approach and previous works is only briefly mentioned, and a single sentence in the Related Work section does not sufficiently clarify the novelty or improvements conveyed by this work. For example, the statement "do not account for SQL generation errors" (Line 110) is vague without examples or tables. It is unclear why SQL generation errors are a critical focus. Do the authors envision text-to-SQL systems returning SQL queries with an explicit “unreliable” label to users? If so, why not fix the errors if they are detectable? Also as examples showing in Figure 1, "reliable" Text-to-SQL systems return an empty response by abstention. In this case, [1], an encoder-based classifier, can also achieve this by translating it from negative labels.

Similarly, the authors argue that template-based methods reduce diversity (as stated in Appendix D), but they also implement templates to structure data, followed by paraphrasing (as in [3]). The lack of a **diversity comparison** between their work and [2] weakens their claim that their methods result in more diverse infeasible question types.  The diversity the authors refer to in Appendix D seems to pertain to linguistic variation, not the diversity of categories of infeasible questions, which considered not quite important. Did this modification lead to some strong contributions of reliability of text-to-SQL systems?  Without experimental proof, it is hard to assert that template-based data generation inherently leads to less diversity. Furthermore, I also think the contribtion is quite limited if the diversity just refers to linguistic difference since [3] proposes much more formats of diversity including lingusitc modification and [1][2] already mention this problem. Also, following [3], please present and append your main types of templates

Given the availability of SPIDER and BIRD, which contain more domains and template-free data annotations, it remains unclear why the authors chose not to leverage these existing resources. These datasets naturally include ambiguous questions, as demonstrated in [4] and [5]. Why not explore them on such existing datasets, but spend more resources in re-annotation?

5) Question Familarity definition and detection is not clear and stable. As we know the same question may have multiple forms of gt SQLs. Clustering by GT SQLs from dataset may not be objective. For example, `highest score` can refer to SQLs with `max()` or `order by limit 1`, would your algorithm consider SQLs with such two keywords as infamiliar questions? This definition is questionable and not intuitive. The same with Question Difficulty in which authors mistakenly assume each question has the only one GT SQL. For example, they consider questions with no JOINs but no nesting in GT SQLs as `Medium`. How about its GT SQL is sub-queried such as `SELECT ... In (SELECT ...)`. It doesn't contain `JOIN` but nesting, what category should belong to? However, it could also be generated as a single `Join`, which should be considered as `Easy`. As pointed in Appendix A in BIRD, the only SQL difficutly cannot represent all difficulties of text-to-SQL tasks. The NL and environment with different complexity of DB schema and constrains should also be considered since it represent "text-to". In a short, I think the whole definitions in the second paragraph of Section 4.3 are not rigorous leading to unfair evaluation setting and distributions especially given authors didn't present a clear distribution and statitic analysis of their benchmark.

### RS Metric:
RS relies heavily on a user-defined penalty factor c. While this allows for flexibility, it also introduces subjectivity. Different penalty values could lead to significantly different conclusions about model reliability, and there's no clear guidance on how to choose an appropriate penalty for a given application. And no appropriate range of c has been discussed. And given this work was positioned as reflections of real-world text-to-SQLs. It would be better disccuss which ranges of c selected can represent which groups of people. For example, whether data analysis with strong SQL knowledge requires just small penalty since they can fix issues if just small alerts are allowed. If this user have high-privacy considerations or totally unaware of SQL knowledge, this penalty could be set to large? Did different range of c will lead to performance shift of LLM or strategy based on GPT-4o?  As a core contribution, the absence of detailed analysis about c's implications and practical application is a significant weakness.

Also, what does \Phi{} mean in Line 351-352, and N/2, this is hard to understand and i didn't find where the big \Phi was defined before. And why implement these three metrics here?

### Experiments:

1) The authors claim to include SQLCoder-2 as the state-of-the-art (SOTA) model. However, several crucial points about this are unclear. First, there are no links, references, or citations to provide more information about SQLCoder-2, and I couldn't locate it on established leaderboards such as SPIDER or BIRD. Additionally, the paper misses details on whether SQLCoder-2 is used as an encoder or decoder-based language model. It would be more appropriate to benchmark against well-recognized models like CodeS [6], which is a widely accepted SOTA model on the BIRD leaderboard. Testing a single model (SQLCoder-2) seems insufficient, and the paper would benefit from comparing performance with other popular large language models (LLMs) like CodeLlama, LLaMA 3/3.1, DeepCoder, and StarCoder. Furthermore, the decision to fine-tune models on the additional dataset TriageSQL, rather than using the training set of TrustSQL, requires more justification. Since this is a benchmark study, it raises concerns about whether other users would also need to fine-tune their models on TriageSQL before proceeding with TrustSQL. Lastly, the mention of "sub-model" is ambiguous—if this is referring to a mixture-of-experts (MoE) system, it should be explicitly stated. Otherwise, it is unclear why a single model's performance is not evaluated. This leaves questions regarding whether the authors are testing text-to-SQL systems as a whole or focusing purely on MoE systems.

2) Another point of concern is the separate implementation of CL methods for SQLCoder-2 and uncertainty estimation (UE)-based methods for T5. It would be more informative to apply both methods to the same model for a more direct comparison, as was done with GPT-4o. If there are specific reasons preventing SQLCoder-2 from being used with UE-based methods or preventing T5 from using CL-based methods, these limitations should be explicitly stated. This would help clarify the generalizability of these techniques. Additionally, the description of the CL-based method implementation is vague, making it difficult to understand how it differs from existing methods like DTE [1]. If CL-based methods are a core contribution of this work, the paper should include a comparison with prior methods and highlight the improvements made in this work.

3) The explanation of how uncertainty estimation (UE) is applied to GPT-4o is unclear. In Line 118, the authors state that the UE-based method can enhance LLM safety by qualifying model confidence, but the connection between this claim and the four dimensions defined in the Related Work section is not sufficiently explained. Moreover, the paper does not clarify how model confidence is represented for GPT-4o, given that it is a closed-source model. This is a crucial point that needs further elaboration to ensure the reproducibility and transparency of the method.




[1] Know what I don’t know: Handling ambiguous and unknown questions for text-to-SQL (wang et al., ACL 2024) \
[2] Ehrsql: A practical text-to-sql benchmark for electronic health records (Lee et al., NeurIPS 2024) \
[3] Dr.Spider: A Diagnostic Evaluation Benchmark towards Text-to-SQL Robustness (Chang et al., ICLR 2023) \
[4] Evaluating cross-domain text-to-sql models and benchmarks (Pourreza et al., EMNLP 2023) \
[5] Understanding the Effects of Noise in Text-to-SQL: An Examination of the BIRD-Bench Benchmark (Wretblad et al., 2024) \
[6] CodeS: Towards Building Open-source Language Models for Text-to-SQL, (Li et al., SIGMOD 2024)

**Questions:**

1) Why did you choose to re-annotate existing datasets with just limited domains like ATIS and EHRSQL rather than leveraging larger benchmarks such as SPIDER and BIRD which contains ambiguous questions covering multiple domains already? What are trade-offs they considered in choosing these specific datasets over more diverse ones like SPIDER or BIRD? Could you explain the specific guidelines used for generating infeasible questions, and how your approach meaningfully improves upon template-based methods like EHR-SQL? Please provide concrete examples that demonstrate how your method yields more diverse or higher-quality data compared to existing techniques.

2) Could you describe the **full annotation process** and **quality control** procedures used in your work? Specifically, how many annotators were involved, what were their expertise levels, how were they recruited, and what compensation was provided? Additionally, did you employ expert panels for review, and what specific measures were implemented to ensure the quality and consistency of annotations across the dataset?

3) How does your system address the complexity of SQL query equivalence in its classification scheme? For example, how are questions handled when they have multiple valid SQL representations (e.g., using `MAX` versus `ORDER BY LIMIT 1`), or when queries can be written both with and without nesting? Furthermore, why weren't factors such as natural language complexity or database schema constraints included in your difficulty assessment? See details in Weakness.

4) Could you provide a quantitative comparison of the diversity and distribution of questions in your dataset relative to existing datasets? This should include specifics on the types of templates, distribution across question categories, and empirical evidence demonstrating how your approach achieves greater diversity. How do these distributions align with the natural distribution of questions in real-world scenarios as authors claimed?

5) In terms of RS metric: What is the recommended range for the penalty factor `c`, and how should it be adjusted for different user groups, such as SQL experts or novices? What empirical evidence supports the choice of these ranges? Additionally, could you clarify the role and definition of large `\Phi{}` parameter and the rationale behind using `N/2` in your formula? How sensitive is model performance to changes in these parameter values? See details in Weakness.

6) What was the reason behind selecting SQLCoder-2 as the primary benchmark model? Can you provide details about its architecture (i.e., whether it is encoder- or decoder-based) and explain why other mainstream large language models (LLMs) were not included in the comparison? How does performance of SQLCoder-2 compare to other SOTA models such as CodeS on standard benchmarks?

7) One crucial point of concern is the separate implementation of CL methods for SQLCoder-2 and uncertainty estimation (UE)-based methods for T5. It would be more informative to apply both methods to the same model for a more direct comparison, as was done with GPT-4o. If there are specific reasons preventing SQLCoder-2 from being used with UE-based methods or preventing T5 from using CL-based methods, these limitations should be explicitly stated. This would help clarify the generalizability of these techniques. Additionally, the description of the CL-based method implementation is vague, making it difficult to understand how it differs from existing methods like DTE . If CL-based methods are a core contribution of this work, the paper should include a comparison with prior methods and highlight the improvements made in this work.

Please see other questions in Weakness for details.

---

> ### Author Response · Authors · 2024-11-27
> **Author Response 1**
>
> Thank you for your detailed comments on our paper, which have been very helpful in improving the clarity of our manuscript. We have restructured your questions and addressed them point by point.
>
> **Q1: Domain generalization concerns and the reason for not using Spider and BIRD**
>
> The inclusion of the three domain-specific databases aims to evaluate whether methodological trends for reliable text-to-SQL modeling remain consistent across different databases, rather than assessing a model's cross-domain generalization ability in SQL generation. Our benchmark focuses on evaluating a model's ability to discern whether to generate SQL or abstain, emphasizing reliability over general SQL generation across varied databases. Unlike benchmarks aimed at domain generalization, our task prioritizes reliability in domain-specific challenges—handling questions that reflect user needs, complex databases, and rich question-SQL assumptions—while incorporating penalties for incorrect decisions. Consequently, high-quality annotations are essential to accurately assess model performance in these specific contexts. To ensure this quality, we ruled out large-scale crowdsourcing and manually curated annotations using three complex domain-specific datasets to maintain the benchmark's integrity.
>
> We also considered using only the development sets of Spider and BIRD but excluded them for the following reasons. Spider is nearly solved as a benchmark, with most errors arising from annotation issues rather than modeling challenges, which limits its usefulness for our reliability-focused evaluation. BIRD's evidence- (or hint-) based text-to-SQL setting is incompatible with our benchmark, as our task requires incorporating both feasible and infeasible questions as model input. For further discussion, please refer to GC2 in the general comment section above.
>
>
> **Q2: More details about the annotation process (number of annotators, expertise level, annotation procedure, ensuring consistency between annotators, quality assurance)**
>
> As summarized in GC3 of the general comments, our annotation process involved three annotators (the authors), all proficient in SQL. We followed a systematic procedure to ensure consistency and quality:
> - Feasible Questions: One annotator reviewed and re-annotated question templates and SQL structures. Two annotators corrected natural language paraphrases. All three annotators met to resolve disagreements and ensure consensus.
> - Infeasible Questions: Annotators used specific keywords to modify feasible questions, creating infeasible ones. Real-time collaboration ensured consistency and high-quality annotations.
>
> For more details, please refer to the revised manuscript.
>
>
> **Q3: The advantage of the proposed infeasible data creation instead of template-based method**
>
> Our keyword-based question creation method combines the strengths of template-based and template-free annotation methods. Annotators are provided with specific keywords and sample feasible questions. They modify these questions to make them infeasible by incorporating the keyword's intent. This approach introduces greater semantic diversity and creates infeasible questions that closely resemble real user questions, enhancing the dataset's realism and the model's ability to handle complex scenarios.
>
> ### Template-based method:
> Consider this infeasible question template: "When is the next earliest hospital visit of patient 0000?"—where no record exists for the next hospital appointment in the database. Possible paraphrases generated from this template can be the following:
> - "What is the soonest upcoming hospital visit scheduled for patient 0000?"
> - "When is the next scheduled appointment for patient 0000?"
> While this method can effectively handle common unanswerable questions (missing-schema), the diversity of the question pool generated using this method may be limited.
>
> ### Keyword-based method (this work):
> Suppose the keyword "appointment" is provided, along with sample feasible questions. The task is to modify these questions to include the keyword, making them infeasible.
> Sampled feasible questions:
> - "Has patient 0000 gotten any medication this year?"
> - "Provide the count of hospital visits for patient 0000."
> Annotated infeasible questions (the infeasible keyword is now inserted):
> - "Has patient 0000 gotten any medication this year and do they have any upcoming appointments?"
> - "Provide the count of hospital visits for patient 0000 including any scheduled upcoming appointments."
> By incorporating the keyword into existing feasible questions to make them infeasible, this method ensures both semantic diversity and guarantees that the annotated questions are indeed infeasible. This approach allows us to create a wider range of infeasible questions that closely resemble real user questions.

---

> ### Author Response · Authors · 2024-11-27
> **Author Response 2**
>
> **Q4: Comparison of data distributions between the authors’ dataset and related work**
>
> We have provided a table summarizing how TrustSQL differs from other benchmarks in GC1 of the general comments section.
>
> **Q5: Definition of query familiarity and query difficulty**
>
> Query familiarity refers to whether a question's original template is included in the training set. Unlike crowdsourced datasets that use template-free data creation, TrustSQL leverages a template-based approach. In the SQL query validation process (detailed in our re-annotation process), we verify that question templates are semantically the same. Given that there are over 180 templates on average per each dataset, assessing pairwise similarity directly is impractical. Therefore, we adopted a two-stage procedure: (1) We grouped question templates that use the same placeholders. For example, "Tell me flights from city_name1 to city_name0" and "What are airlines that provide services from city_name1 to city_name0" share the placeholders city_name1 and city_name0; (2) Within each group, we reviewed the templates to determine if they are semantically identical. If they were, we merged the templates and paired them with a unique SQL structure. This process allows us to maintain control over the semantic distribution of questions across data splits.
>
> Regarding the classification of query difficulty, we acknowledge that SQL complexity alone does not encompass all challenges in text-to-SQL tasks. Nonetheless, we believe SQL complexity remains the most significant factor in assessing query difficulty, as it directly reflects the complexity of the model's expected output. While BIRD, as the reviewer mentioned, introduces four dimensions to measure sample difficulty, these guidelines are inherently subjective and may vary among annotators. For instance, their annotation framework rates 1 as a simple SQL with few keywords, 2 as more complex than 1, and 3 as a highly complex SQL with many functions, which can result in inconsistencies across samples. Similarly, while DIN-SQL's approach is not entirely rigorous, it is straightforward and demonstrates strong correlations with model performance—as query difficulty increases, performance decreases (see Appendix B.2 on SQL generation).
>
>
> **Q6: Diversity and distribution of questions**
>
> We utilize existing dataset question templates, so the types and distribution of templates reflect those in the original datasets. The number of question templates for ATIS, Advising, and EHRSQL is 238, 141, and 168, respectively. In our paper, the term "diverse input" refers to a setting where the text-to-SQL model is tasked to process various types of questions, including both feasible and infeasible ones.
>
> **Q7: The selection of c**
>
> We have elaborated on the selection of c in the general comments section above (see GC4).
>
>
> **Q8: The choice of fine-tuned models (SQLCoder-2)**
>
> SQLCoder-2 (defog/sqlcoder-7b-2) is an open-source SQL-specialized model based on CodeLlama 7B. Although it may not be widely used in cross-domain generalization benchmarks, this was not a key consideration, as we do not conduct cross-domain tasks in our experiments. SQLCoder-2 was chosen because it outperformed the fine-tuned Llama 3.1 8B in our initial tests. We have clarified in the manuscript that SQLCoder-2 is one option for a decoder-only model, whereas T5-3B is an encoder-decoder model.
>
> **Q9: Compatibility of abstention mechanisms across fine-tuned models**
>
> SQLCoder-2 is a decoder-only model, while T5 is an encoder-decoder model. Their respective abstention methods may not always be compatible. We chose these to demonstrate how different architectures can be paired with compatible abstention mechanisms in various ways.

---

> ### Author Response · Authors · 2024-11-27
> **Author Response 3**
>
> **Q10: Rationale behind using TriageSQL for training infeasible question classifiers (INF+)**
>
> Our dataset does not include infeasible questions in the training set, and TriageSQL is the only openly available dataset for question classification in database question answering. Therefore, we used it to train the infeasible question classifier. In task-oriented dialog systems, out-of-intent questions (i.e., infeasible questions) are typically excluded from the training set [1,2]. Meanwhile, using external datasets is common in the era of pre-trained models. For those who prefer not to use TriageSQL to fine-tune their models, in-context learning is a viable alternative, as demonstrated in one of our GPT-4o-based baselines.
>
> [1] Zheng, Yinhe, Guanyi Chen, and Minlie Huang. "Out-of-domain detection for natural language understanding in dialog systems." IEEE/ACM Transactions on Audio, Speech, and Language Processing 2020.
> [2] Marek, Petr, Vishal Ishwar Naik, Anuj Goyal, and Vincent Auvray. "OodGAN: Generative Adversarial Network for Out-of-Domain Data Generation." NAACL 2021 Industry track.
>
> **Q11: The term sub-model**
>
> Sub-models refer to the individual models in the CL-based text-to-SQL pipeline. To clarify, we have updated the manuscript: "These models operate sequentially, with task-specific sub-models functioning in the following order: infeasible question detection, SQL generation, and SQL error detection."
>
> **Q12: Meaning of big \Phi**
>
> Φ is defined in Formula (1).

---

> > ### Comment · Reviewer_xD5x · 2024-11-28
> > **Thanks for answering my questions**
> >
> > Thank you for your efforts in addressing my concerns and providing responses to my questions. However, after carefully reviewing the revised PDF and the responses, I still believe there are significant issues related to potential bias in the dataset, and I remain unsatisfied with the explanations provided. I will keep my original score based on the following points:
> >
> > 1) Annotation Process and Bias Concerns: the description of the annotation process remains unclear, and without details on **inter-annotator agreement** and a **well-defined taxonomy**, it is difficult to be confident that the dataset is free from bias. Authors emphasize terms like "verify," "ensure quality," and "guarantees," but these are not supported by any **quantitative evidence** or clear explanations of how. How do you verify the quality or ensure diversity in the annotations? Showing concrete numbers (e.g., inter-annotator agreement scores), user studies or examples of ambiguity types would make your claims more convincing.
> >
> > Additionally, as mentioned in my original comments, I believe the paper should present a clear taxonomy of infeasible question types, along the lines of related works such as [1], to illustrate the range of question types and their distribution in the dataset. Without such a taxonomy, I cannot trust that the benchmark does not exhibit bias. For example, using the taxonomy from Table 1 in [1], can you demonstrate **whether all types of infeasible questions in your dataset are only due to column ambiguity**? If not, how do you justify this in an academic way?
> >
> > Also, authors mention that the human-rated difficulty in BIRD is quite subjective. Similarly, how were the keywords selected in this paper, and how was annotator agreement ensured, especially when the **annotators are also authors of this paper and not independent experts**? I even think the promised "ensure" would be more subjective since annotators in the work are also authors. This concern has to be solved as a benchmark.
> >
> > 3. **Contributions and Comparison with Existing Works**: The paper requires a deep understanding of existing literature, and the presentation in Table GC1 does not adequately compare your work to others. Typically, such comparisons are made to reflect the **unique contributions** of a new paper. In this case, since your focus is on ambiguous and unanswerable questions, it is important to demonstrate how your work adds to or differs from existing studies like DTE.
> >
> > I don't think it's useful to show comparisons with other features. For eg., what is point to show  longer SQL outputs and larger schemas are associated with more infeasible questions? **Is there a proven correlation between SQL output length, schema size, and question infeasibility?** If so, this should be shown clearly by references, or your study. Otherwise, Table 1 seems not insightful, as it merely reiterates findings from DTE without offering novel insights in terms of infeasible questions.
> >
> > 4. Not Clear Understanding of **SOTA** Models and Benchmarking:
> > It is important to clarify what "SOTA" (State-Of-The-Art) means in this context. In the text-to-SQL field, SOTA typically refers to models that achieve top rankings in widely recognized benchmarks, such as LGESQL, RATSQL, CodeS, Graphix-T5, DIN-SQLs in SPIDER.  All of them are also open-source, why just conduct one decoder-only model, which is not popular in the experiment? The model you cite, "defog/sqlcoder-7b-2," has not been demonstrated to outperform SOTA models in these established benchmarks. The fact that "defog/sqlcoder-7b-2" is not included in these widely recognized benchmarks raises questions about what is  "SOTA". More importantly, it is crucial to note that the performance of SQLcoder-7b-2 was trained and evaluated in **PostgreSQL-based** settings, while your dataset is based on **SQLite**. Different SQL dialects have different syntax and constraints. How can you justify employing this model which is good at PostgreSQL also good at SQLite?
> > if so, provide evidence supporting it's better than any of SOTA models in SQLite-based benchmarks such as CodeS.
> >
> > 5. Diversity of Infeasible Questions: Table 2 shows simple clarifications of question types, why not add comparisons with existing works, such as DTE or EHR-SQL, to highlight what is new and unique in your benchmark? For example, **what types of infeasible questions in your dataset are distinct from those in DTE or EHR-SQL?** I could not find such a comparison in the revised paper. This would be important not only to demonstrate the novelty of your work but also to clarify how your benchmark contributes to advancing the field. Otherwise, if existing work already contains all your types of infeasible question types, then what is the point of this research? Please learn how to conduct taxonomy presentation by [1] and Figure 8 in BIRD.

---

> > > ### Comment · Reviewer_xD5x · 2024-11-28
> > > **Comments 2**
> > >
> > > 6. Choice of Metric (c): I understand that this work is focused on dataset construction and benchmarking. However, the metric you introduce seems to be a specific contribution compared to related work, but it is not analyzed in depth, and its significance is unclear. The lack of a thorough explanation raises concerns about the fairness of the entire benchmark and reproducity. Without a clear understanding of how the metric was developed, validated, and its implications for benchmarking, it is difficult to fully judge its utility. Without this contribution, I have no idea what is special compared to [1] and EHR-SQL.
> > >
> > > 7. Domain Bias:
> > > Authors mention that the influence of domain diversity on infeasible questions is not a primary focus of your paper. However, I question why you chose to include datasets from three different domains. If domain diversity is not a key consideration, then why not include just one domain? This seems contradictory to your stated goals.
> > >
> > > Moreover, my new question is: Do the types of infeasible questions predominantly arise from biases inherent to specific domains (e.g., the clinical scenarios you mention)? If so, does this mean the types of infeasible questions you identify are specific to certain domains and not generalizable across all text-to-SQL tasks?

---

> ### Author Response · Authors · 2024-12-01
> **Author Response 1**
>
> Thank you for the detailed review of our paper and for engaging in the discussion with us. Please find our response below.
>
> **Q1: Annotation quality concerns**
>
> Regarding concerns about annotation quality, TrustSQL consists of three components: **feasible data**, **infeasible data**, and corresponding **databases**. First, the **databases** we use are openly available and feature complex schemas compared to others in the text-to-SQL literature (see comparison in GC1), which are far from the bias you mentioned. Second, the **feasible data** include questions, SQL queries, and evidence. These questions and SQL queries are sourced from existing datasets—we are not their original creators but act as reviewers and correctors. This approach is more like a cross-project review/annotation process rather than a single-project effort, which helps mitigate concerns about annotation bias compared to other works. Lastly, for **infeasible data**, we introduce infeasible keywords (detailed in Appendix A.3.1–A.3.3) and incorporate them into feasible questions to make them infeasible. Below, we summarize our annotation process for infeasible questions (detailed in Appendix A.3) and explain how we ensure that adding infeasible keywords consistently results in infeasible questions.
>
> ### Missing-schema
> Infeasible questions belonging to the missing-schema category are annotated based on hypothetical columns (columns that do not exist in the actual databases) listed in Appendix A.3.1. These columns are designed to mislead the model, and they are uniquely written for each table. For example, we create an infeasible question using one of the infeasible keywords “NUM_LOUNGE” in the airport table from ATIS. Starting with a feasible question like “What are the flights that leave from DAL and go to other airports?,” the annotators modify it by incorporating the meaning of the keyword to form a natural question, such as “Find the average number of lounges across all airports.” If the annotated question correctly references these keywords, we can ensure that it is infeasible because the referenced columns do not exist in the actual database.
>
> ### Ambiguous and Non-SQL
> The keywords in these categories are not database-specific but are instead general keywords that make questions infeasible. For ambiguous questions, given a list of feasible questions, the annotators are tasked with inserting vague words (e.g., “suitable,” “best”) or referentially ambiguous terms (“those,” “this”) (details in Appendix A.3.2). For non-SQL questions, task-related keywords (e.g., “clustering,” “sentiment analysis”) are provided, and the annotators modify questions by incorporating these keywords naturally (details in Appendix A.3.3).
>
> After all, the quality check involves verifying whether the keywords are truly infeasible and ensuring that the annotated questions properly reflect these keywords.
>
> Regarding the full taxonomy, we initially included Table 2 to provide a brief overview of the categories of infeasible questions, with more detailed taxonomies for each type presented in Table 8 (ambiguous) and Table 9 (non-SQL). However, we plan to include a unified and comprehensive taxonomy of question types in the revised manuscript.
>
>
> **Q2: Contributions and comparison with existing works**
>
> As the title of the paper suggests, our work aims to measure “text-to-SQL reliability.” To evaluate reliability in more realistic scenarios, we consider complex databases (↑ schema size), questions requiring complex SQL queries (↑ #Tab/DB and ↑ Avg SQL Tok), and a new task setting where user questions include both feasible (AnserQ) and infeasible (UnansQ and AmbigQ) cases. Additionally, we introduce an evaluation penalty (Eval Penalty) to address varying user safety requirements. At the core of our work is answering this critical question: **”Is my model reliable enough for safe deployment compared to others?”**—quantified by calculating the difference between helpfulness (i.e., the number of correct model decisions) and harmfulness (i.e., the number of incorrect decisions weighted by a penalty). No existing single work directly addresses this question in the context of safe text-to-SQL model deployment.
>
> **Q3: The choice of SQLCoder**
>
> Thanks to your suggestion, we have avoided using the term 'SOTA' when referring to SQLCoder in the revised manuscript. Instead, we describe it as a decoder-only, SQL-specialized Code Llama. Regarding your comment on SQLCoder's PostgreSQL setting, we fine-tune SQLCoder as a decoder-only model on the training portion of TrustSQL, so its original SQL dialect is not a major concern for us. Of course, using CodeS or other models could be another option for fine-tuning a decoder-only model, and it might perform better on SQL generation. However, we believe this change would not significantly affect the claims made in our paper. What contributes most to this benchmark is the use of abstention strategies, especially as the penalty increases.

---

> ### Author Response · Authors · 2024-12-01
> **Author Response 2**
>
> **Q4: Diversity of infeasible question and novelty of this work**
>
> Thanks to your suggestion, we have included a comparison between keyword-based and template-based (EHR-SQL) infeasible data generation in Appendix A.4. While extensive coverage of infeasible questions is valuable, we argue that it does not fundamentally alter our main claim. Our primary focus is to evaluate the reliability of text-to-SQL models in answering the question, “Is my text-to-SQL model the best and reliable enough for safe deployment compared to others?”—rather than simply classifying input question types, as done by TriageSQL and DTE. Infeasible questions are incorporated to simulate a more realistic evaluation setting, where not all questions are feasible, and we annotate these questions based on common types observed in previous user studies.
>
> Since you question the diversity of infeasible questions compared to other works, we provide a more detailed discussion below:
>
> TriageSQL’s infeasible question types:
> - Improper: Random utterances.
> - ExtKnow: Questions that cannot be answered using the given schema.
> - Ambiguous: Ambiguity in column references.
> - Non-SQL: Operations beyond SQL’s scope.
>
> DTE’s infeasible question types:
> - Column ambiguity: Ambiguity in column references.
> - Column unanswerable: Questions that cannot be answered using the given schema.
> - Additional types mentioned in DTE, such as value ambiguity, value unanswerable, calculation unanswerable (requiring external knowledge of formulas), and out-of-scope questions (beyond SQL’s scope), are excluded in DTE as they are found to be less frequent in their user study.
>
> Infeasible question types in our benchmark compared to TriageSQL and DTE:
> - Missing-schema: Corresponds to ExtKnow and Ambiguous in TriageSQL and to column unanswerable and ambiguity in DTE.
> - Ambiguous: Covers referential ambiguity and vagueness*. While DTE and TriageSQL limit ambiguity to column-related issues, we include different types but adhere to the same principle: questions cannot be feasible without further clarification.
> - Non-SQL: Matches the out-of-scope category in DTE and the non-SQL category in TriageSQL.
>
> Categories not included in our benchmark and reasons for exclusion:
> - Improper (TriageSQL): Our preliminary analysis showed that random utterances sampled from other datasets are too easily filtered out to add as a meaningful category.
> - Value ambiguity/unanswerable (DTE): Determining whether a specific value exists in the database exceeds the scope of a single-turn text-to-SQL setting.
> - Calculation unanswerable (DTE): With models like GPT-4o already having extensive knowledge of formulas, labeling such questions as unanswerable is no longer appropriate.
>
> \* One fundamental cause of infeasible questions is the ambiguity that arises from the richness of natural language expressions and the habitual omission of details by users [1, 2].
>
> **Q5: Choice of Metric (c)**
>
> As the title of our paper suggests, penalty-based scoring is a key contribution of our work. Existing text-to-SQL approaches do not penalize incorrect outputs, thereby overlooking the importance of model abstention in model development. By introducing this metric into text-to-SQL evaluation, we provide a way to measure user-defined reliability—specifically, whether the model's helpfulness outweighs its harmlessness—an aspect that no other text-to-SQL benchmark currently offers.
>
> To answer the question "how the metric was developed," we adapt a metric originally proposed in Reliable VQA [3] for text-to-SQL tasks and extend it to address unanswerable questions. To answer "how the metric was validated," our experiments demonstrate how the reliability score (RS) correlates with coverage and risk, making it a practical tool for selecting models under different user-defined safety requirements. Regarding "its implications for benchmarking," we observed that models like GPT-4o perform well with low penalties, while T5-based models with threshold mechanisms excel in high-penalty settings due to their conservative prediction strategies—findings that would not have been discovered without this benchmark. We believe this contribution is essential for the adoption of text-to-SQL models in real-world industries with varying safety needs.
>
> [1] Wang et al., "Know what I don’t know: Handling ambiguous and unknown questions for text-to-sql." ACL Findings 2023.
> [2] Radhakrishnan et al., "ColloQL: Robust text-to-SQL over search queries." IntEx-SemPar 2020.
> [3] Whitehead et al., “Reliable Visual Question Answering: Abstain Rather Than Answer Incorrectly.” ECCV 2022.

---

> ### Author Response · Authors · 2024-12-01
> **Author Response 3**
>
> **Q6: Domain bias**
>
> Our benchmark includes three significantly distinct database domains: airline travel, education, and healthcare. The goal of our work is to evaluate the reliability of models on complex databases and SQL queries, particularly in scenarios where not all questions are feasible. We believe that as long as methodological trends remain consistent across domains, this level of domain diversity is sufficient to validate our claims and conclusions. Indeed, we observed consistent modeling trends across databases in our experiments.
>
> To clarify, our work does not include cross-domain text-to-SQL experiments where no in-domain training data is provided and the number of domains may be a primary concern. As noted in prior work, "the zero-shot setting is overly restrictive compared to how text-to-SQL systems are likely to be used in practice [4]." Similarly, we argue that achieving high performance in the current cross-domain text-to-SQL generalization setting is not the only challenge in this field. Instead, our goal is to develop or select the best models that meet a certain safety standard for text-to-SQL systems with some in-domain data available, which is a more realistic setting in practice.
>
> Regarding domain biases in infeasible questions, we consider only the "missing-schema" type (Appendix A.3.1) to be database-specific, while ambiguous and non-SQL questions are not, as they are annotated with general, database-independent keywords (Appendix A.3.2-3). However, we believe this does not impact the models' behavior across databases. None of the models in our experiments are manually tailored to specific domains. We use the same model architecture and learning algorithm across all three databases, and the conclusions remain consistent.
>
> If you still consider domain bias a concern in our work, we welcome your input on the number and variety of domains needed to sufficiently support our claims.
>
> [4] Lee et al., "KaggleDBQA: Realistic Evaluation of Text-to-SQL Parsers." ACL 2021.

---

> > ### Comment · Reviewer_xD5x · 2024-12-01
> > **Feedbacks of Responses**
> >
> > First, I would like to acknowledge the authors' efforts and their responses to the reviewer's comments. However, I think very few points that I raised that are answered. Some of the responses raised additional concerns that I believe need further clarification.
> >
> > **Q1:** \
> > Authors argue that the quality and absence of bias in the dataset are ensured since it is composed of existing datasets. However, I would like to highlight two key issues:
> >
> > 1. **Upper Bound of Dataset Quality**: Since the dataset is derived from the three existing datasets, its quality is inherently limited by the quality of these sources. Authors cannot resolve the problems present in earlier datasets.
> >
> > 2. **Post-Processing and Data Quality**: The authors mention that they reviewed and modified the data, but there is no clear evidence provided regarding the quality of the **final dataset**, which is presented to the community. It would be crucial to demonstrate the quality of the revised data itself.
> >
> > **The Choice of Infeasible Keywords:** concerns of this remain as I asked last time:
> >
> > 1. Authors have not fully addressed the **criteria** for selecting keywords. What is the guiding principle for this choice? How can the authors assure readers that the keyword selection is not subjective? This is an important issue especially when the annotators are the same as the authors.
> >
> > 2. The concept of "general" keywords is not clearly defined. Is there any metric used to assess the generality of keywords, such as frequency in ambiguous questions within the study? Or is it based purely on subjective feeling? While I appreciate the additional examples provided, I still find it unclear what logic under the example. A more structured summary of these principles would help clarify the reasoning.
> >
> > 3. A recurring concern throughout the review process is no clarity on how the authors performed **quality checks** on their data. What objective metrics or post-evaluations were used? Without clear, measurable criteria, I find it difficult to trust the authors' claims regarding data quality. Please note more metrics, number such as "inter-annotator agreement" are more powerful and convincing than just NL descriptions. Please read more related benchmark papers carefully to understand.
> >
> > 4. I appreciate the authors for striving to make the paper clearer, but there are still several issues regarding the presentation of category distributions. For example, the "Reason" labels are insufficient for conveying the logic behind category choices. Do authors expect readers only understand them by further summarizing behind examples and reasons? The authors should provide both category names and detailed definitions, along with statistical distributions (i.e., portions, percentages) to give readers a more complete understanding. This is a standard approach in the field, as demonstrated in [1], BIRD, which the authors could benefit from reviewing (in which I suggested). Including these distributions will not only make the paper more systematic but also enhance the credibility of the results. Please be aware that "we believe" is no power in academic writing. Readers and reviewers only trust evidence reflected by analysis with plots, numbers, etc.
> >
> > **Q2:** \
> > The use of the terms "reliability" and "safety" is somewhat confusing. The authors first mention "reliability" and later refer to "safety," but the connection between these two terms is not clear. In the paper, "reliability" seems to refer to ambiguous and "non-SQL" behavior, but what exactly is the relationship between this and "safety"? Typically, "safety" would concern preventing privacy violations or defending against adversarial attacks. Such a statement is quite confusing.
> >
> > **Q3:** \
> > I am still unclear about the decision to evaluate a model optimized for PostgreSQL when the dataset is based on SQLite. Given the **wide availability** of advanced text-to-**SQLite** models, why did the authors choose a less well-known **PostgreSQL** model for evaluation? The argument that fine-tuning will resolve the issue seems not trustworthy without evidence. I would appreciate some concrete evidence to support this claim. Otherwise, I will have other concerns. For example, the volume of data may influence. For example, more data is required to eliminate the syntax bias of PostgreSQL.  Furthermore, the statement that "this change would not significantly affect the claims made in our paper" is not a valid argument. I believe that such a significant model choice deserves more than just a promise; clear evidence is needed to make this claim.

---

> > > ### Comment · Reviewer_xD5x · 2024-12-01
> > > **Other Feedbacks**
> > >
> > > **Q4:**
> > > 1. **Input Question Types**: Authors should not overlook the importance of **classifying input question types**. I quite disagree with such an academic attitude, especially by using **simply**. This is a fundamental aspect of ensuring the comprehensiveness and faithfulness of the benchmark benchmark work. Even simple and basic, did authors do that well?
> > >
> > > 2. **Taxonomy of Infeasible Questions**: The authors' attempt to classify infeasible questions is appreciated, but it seems to be limited. In particular, I noticed that:
> > >    - The categories of infeasible questions are already covered by existing works.
> > >    - Most notably, I strongly disagree with the reason of exclusion of **value and calculation**-based questions. These types of questions are common and realistic in text-to-SQL tasks since authors stress on **realistic** along the paper and rebuttal. For example, BIRD, a widely recognized benchmark, explicitly includes values as a key motivation. Similarly, in the work that author cited, KaggleDBQA, they also consider the values of databases are important components. Excluding such questions significantly narrows the scope of the work. The authors argue that value-based questions exceed the scope of a single-turn text-to-SQL setting, but I find this argument unconvincing. All BIRD and KaggleDBQA are single-turn text-to-SQL benchmarks, right? In my view, such reasons are not convincing.

---

> > > > ### Comment · Reviewer_xD5x · 2024-12-01
> > > > **Other Feedbacks**
> > > >
> > > > **Q5:** \
> > > > The penalty-based scoring is quite nice and much valuable compared to other related works. I acknowledged this in my first and last review. All reviewers can see this and we agree this is main contribution. However, such a score seems not stable and can be regulated. Our concern is what is the normal and available range of this metric. How to select or adjust such score to feed users' different needs of their products and models. As you said, GPT-4 and T5 present different performance scope in different choice of penalty scores. Then what is the instruction? What is the evidence to show it's reliable in a certain range? It's quite mysterious. This is not only my concern but also other reviewers.
> > > >
> > > > ### **Domain Bias**
> > > >
> > > > 1. **Clarification of "Cross-Domain"**: What exactly does "cross-domain" mean in your work? Your dataset includes three domains, does this refer to a "cross-domain" dataset, or is it considered a single-domain dataset? If a dataset with three domains is considered "cross-domain," then how do you reconcile this with your statement that "our work does not include cross-domain text-to-SQL experiments where no in-domain training data is provided, and the number of domains may be a primary concern"? This seems somewhat contradictory, and further clarification is needed.
> > > >
> > > > 2. **Consistency Across Domains**: The authors mention, "Indeed, we observed consistent modeling trends across databases." Is this conclusion based solely on the three domains in your dataset? For comparison, KaggleDBQA, which you also cite, includes data from 8 domains. My concern here is with **data bias**, not **model performance**. The fact that you observed consistent model performance across your three domains does not necessarily imply that the dataset is free from bias since it maybe caused by the robustness of LLM itself, not the **dataset** itself.
> > > >
> > > > 3. **Zero-Shot Setting Argument**: You state, "The zero-shot setting is overly restrictive compared to how text-to-SQL systems are likely to be used in practice [4]." This conclusion was made at the time (**2021**) before the ChatGPT era (starting at **Dec, 2022**), the landscape has shifted significantly since the advent of models like GPT-4 and even SQL-Coder, which are capable of strong **zero-shot inference** by simple prompting. It is important to reconsider the relevance of this argument in **2025**. Specifically, my concern is with the evaluation set. How this training is performed (e.g., through in-context learning or fine-tuning), should be part of the challengers' approach. If your benchmark mandates training on a specific training set, this could further limit its usefulness, since challengers should be likely to adopt a range of methods (e.g., zero-shot, fine-tuning) in practice.

---

> > > > > ### Comment · Reviewer_xD5x · 2024-12-01
> > > > > **Other Suggestions**
> > > > >
> > > > > I noticed that authors always try to argue by **"we believe"** along the discussions. I appreciate your opinions, but this word seems too weak in academic discussion or writing. More welcome way is to show evidence to support what "you believe". Please note, you are writing a paper and building a benchmark for public, for readers to accept. Please show more objective analysis with evidences to illustrate them. I think this is more convincing to accept your statement more than just subjective arguments by stating what you "believe".  Thanks for your responses.

---

> > > > > > ### Author Response · Authors · 2024-12-03
> > > > > > **Author Response 1**
> > > > > >
> > > > > > We appreciate your time and engagement in this discussion. By using the word 'believe,' we aimed to share our perspective, recognizing the limitations in providing extensive details or additional experiments within the discussion period. We kindly ask for your understanding of this constraint during the discussion.
> > > > > >
> > > > > > **Q1: Annotation quality concerns**
> > > > > >
> > > > > > 1. Upper Bound of Dataset Quality: We do not agree with your claim that the quality of data is inherently limited by the quality of the sources. The most important quality aspect in text-to-SQL data is the quality SQL annotation with respect to their natural language questions. We have seen groups of researchers like [1, 2] improve the quality of existing datasets.
> > > > > >
> > > > > > 2. Post-Processing and Data Quality: We explain common annotation errors found in existing datasets and describe our process to address these issues in Appendix A.2. This demonstrates clear evidence of data quality improvement. If you have any suggestions for better ways to present the 'quality' of the data, we welcome your input.
> > > > > >
> > > > > > The Choice of Infeasible Keywords:
> > > > > > 1. Below are more details on the keyword creation process:
> > > > > > Keywords for missing-schema: We use GPT-4o to generate additional columns that may be semantically suitable or have similar surface forms to existing columns for each table. When these columns are not diverse enough, the authors create additional columns manually. These keywords are listed in A.3.1., but we plan to provide the full database schema for readers to compare their relevance. Semantically suitable keywords resemble the unanswerable question generation process in TriageSQL and DTE, while similar surface forms resemble the ambiguous column generation process in DTE.
> > > > > > Keywords for ambiguity: We use GPT-4o to generate words that involve referential ambiguity (e.g., “those,” “this”) and vagueness (e.g., “suitable,” “best”). Annotators are provided with sampled keywords, and if the sampled feasible questions are not well-aligned with the provided keywords, the annotators are free to suggest their own keywords, as long as they are referentially ambiguous or vague. This approach reflects the richness of natural language expressions and the habitual omission of details by users.
> > > > > > Keywords for non-SQL (e.g., “clustering,” “data visualization”): These task names reflect questions that users unfamiliar with SQL functionalities may ask database question-answering systems. They often involve machine learning or data science-related tasks that cannot be executed using SQL.
> > > > > > 2. Keywords for ambiguous and non-SQL categories are non-domain/database-specific keywords, while keywords for missing-schema are considered domain/database-specific.
> > > > > > 3. "Inter-annotator agreement" in text-to-SQL data annotation
> > > > > > For annotations requiring high precision, such as SQL annotations, inter-annotator agreement is typically not reported. Instead, annotators work collaboratively to ensure the highest possible quality. Any discrepancies in annotations are resolved to ensure data quality. For instance, in the BIRD dataset, if annotated SQL queries produce different results, "SQLs are checked with experts until a consensus is reached." Similarly, datasets like Spider and KaggleDBQA do not report inter-annotator agreement.
> > > > > > 4. For the proportion of infeasible questions, questions fall into missing-schema, ambiguous, and non-SQL cases are evenly distributed (33% each). As for the taxonomy, we will include it in the revised manuscript.
> > > > > >
> > > > > > **Q2: Relationship between reliability and safety**
> > > > > >
> > > > > > In our paper, we define reliability using the Reliability Score (RS), which quantifies the difference between a model's helpfulness and harmfulness. As shown in Formula (1) for calculating the RS, harmfulness (-c) arises from generating incorrect SQL (e.g., failing to filter out infeasible questions and thus producing incorrect SQL, or failing to abstain from generating incorrect SQL for feasible questions). The term "safety requirement" (or "safety standards/levels") refers to the user-defined penalty, c. A model is said to meet the safety requirement of c if it achieves a positive RS(c), indicating that its helpfulness outweighs its harmfulness under the given safety requirement.
> > > > > >
> > > > > > [1] Wretblad et al., Understanding the Effects of Noise in Text-to-SQL: An Examination of the BIRD-Bench Benchmark. ACL 2024.
> > > > > > [2] Finegan-Dollak et al., Improving Text-to-SQL Evaluation Methodology. ACL 2018.

---

> > > > > > > ### Author Response · Authors · 2024-12-03
> > > > > > > **Author Response 2**
> > > > > > >
> > > > > > > **Q3: The choice of SQLCoder**
> > > > > > > SQLite and PostgreSQL share significant overlap in SQL syntax, which is why we chose to fine-tune SQLCoder instead of Llama 3.1 8B. Furthermore, we view SQLCoder to be far from a “less well-known” model. On Hugging Face, SQLCoder-7B-2 recorded 49K downloads last month, whereas CodeS, the model you referenced, had fewer than 5K downloads across all its parameters and variations combined, as of this writing. Additionally, one of the most widely used open-source LLM deployment libraries, Ollama, actively hosts SQLCoder (https://ollama.com/search). Which of our claims do you think would change significantly if we replaced the fine-tuned SQLCoder with another fine-tunable LLM? We intentionally use the word 'believe' to express our opinion, whereas you assert that our claim is 'not valid' as if it were an absolute truth.
> > > > > > >
> > > > > > > **Q4**:
> > > > > > > 1. We never overlooked the importance of classifying input question types. Classifying questions prior to SQL generation is a fundamental aspect to uphold the assumption of the current text-to-SQL modeling. Knowing this importance, we include this very aspect in our task formulation and baseline evaluation. The term "simply" was used to emphasize the relative complexity of the task. Our definition of measuring reliability involves not only filtering infeasible questions, but also SQL generation and detecting errors in the generated SQLs.
> > > > > > >
> > > > > > > 2. As mentioned in our paper, the primary reason for excluding these categories is that they are not among the most common infeasible types identified in prior user studies. Furthermore, we believe their exclusion does not “significantly” narrow the scope of our work. To the best of our knowledge, none of the existing question detection works explicitly address these categories. Regarding the term “realistic,” our focus is on addressing the most frequent and problematic cases, rather than attempting to cover every possible scenario. If the current three infeasible categories are not properly filtered, it is unlikely that a system would be considered more reliable than models that succeed in filtering other categories but fail on these. From a technical perspective, as mentioned in our previous comment, the “calculation unanswerable” category is no longer infeasible, as models like GPT-4o can handle such questions, making them feasible. Similarly, “value ambiguity/unanswerable” becomes feasible when there is documentation or evidence of value references in the database, as demonstrated in datasets like BIRD and KaggleDBQA. However, without such references, which is often the case since documenting all possible variations of values in natural language questions beforehand is impractical, determining whether a specific value exists in the database often requires multi-turn interaction with the user.
> > > > > > >
> > > > > > > **Q5:**
> > > > > > > It is unreasonable to expect us to define a specific value of c for every domain or task. This decision should be made by experts with deep domain knowledge, tailored to their specific deployment circumstances. We have already provided sufficient high-level guidance in our paper: for less safety-critical settings, we suggest using 1 (or even 0), and for safety-critical settings, N (or even -inf).
> > > > > > >
> > > > > > > **Q6:**
> > > > > > > 1. Cross-domain/database and single-domain/database are experimental settings [3,4], not the number of domains within one dataset. Single-domain focuses on a specific database, optimizing with its schema and in-domain question-SQL pairs for high accuracy. Cross-domain uses diverse databases for training, testing on unfamiliar ones. This work focuses on single-domain settings.
> > > > > > > 2. We never claimed that this dataset is completely free from domain bias. What we emphasized is that as long as consistent modeling trends are observed across databases using widely recognized baselines like T5, Code Llama variant, and GPT-4o, none of which are specifically designed for robustness, adding more databases was not necessary to validate our claim.
> > > > > > > 3. The statement “The zero-shot setting is overly restrictive compared to how text-to-SQL systems are likely to be used in practice” is still valid in today’s high-risk deployment settings. If text-to-SQL is used for an experimental purpose, zero-shot inference is permissible. However, if the consequence of an incorrect answer is high, zero-shot inference would not be used for deployment. We provide details about model training in Appendix C. For the GPT-4o baselines, no training is involved; instead, in-context learning is used.
> > > > > > >
> > > > > > > [3] Chang and Fosler-Lussier. How to Prompt LLMs for Text-to-SQL: A Study in Zero-shot, Single-domain, and Cross-domain Settings. TRL Workshop 2023.
> > > > > > > [4] Suhr et al., Exploring Unexplored Generalization Challenges for Cross-Database Semantic Parsing. ACL 2020.

---

> > > > > > > > ### Comment · Reviewer_xD5x · 2024-12-03
> > > > > > > > **Disagree with Usage of PostgreSQL model for SQlite dataset**
> > > > > > > >
> > > > > > > > I disagree the metrics of downloads in hf can be considered as academic metrics in benchmark. Did authors count and study whether more downloads of Defog for products or SQLite evaluation? Have author proposed any papers doing the same thing? I highly keep my concerns about the faithfulness of the dataset due to such huge bias and not objective view of this work.

---

> > > > > > > > ### Comment · Reviewer_xD5x · 2024-12-03
> > > > > > > > **Keep concerns for data quality problem**
> > > > > > > >
> > > > > > > > To the end, authors still not answer my questions. And I just suggest authors read more papers to compare what should do for others trust your statement. At least for now, any interpretations are not clear to make me trust its diversity, quality without bias. Using existing dataset to assume post-processed quality of data is quite unprofessional!
> > > > > > > >
> > > > > > > > For choice of c: without any detailed analysis about it, we rather than consider it as an unstable factors of benchmark. For example, please refer to how detailed test suite in Spider, VES in BIRD. I think when presenting a new metric, the detailed discussion including range, instructions of how to regulate or use it is quite basic! I don’t see any places unreasonable.

---

> > > > > > > > > ### Author Response · Authors · 2024-12-03
> > > > > > > > > **Author Response**
> > > > > > > > >
> > > > > > > > > We appreciate your enthusiasm to engage with us and assert your position on our work. We will take what we can from this discussion to improve our work in the future.

---

### Official Review · Reviewer_sco7 · 2024-10-30

**Soundness:** 3
**Presentation:** 3
**Contribution:** 2
**Rating:** 5
**Confidence:** 4

**Summary:**

This paper introduces TrustSQL, a benchmark aiming to evaluate the reliability of text-to-SQL models in handling both feasible and infeasible natural language questions. The authors identify limitations in existing benchmarks, specifically the lack of infeasible questions and abstention mechanisms in models, which can lead to incorrect or harmful responses. They propose the Reliability Score (RS) metric to quantify a model's helpfulness relative to its potential harmfulness, assigning positive scores for correct decisions and penalizing incorrect ones based on a user-defined penalty. The authors re-annotate three domain-specific datasets—ATIS, Advising, and EHRSQL—to include infeasible questions categorized into missing schema, ambiguous, and non-SQL types. Various text-to-SQL models integrated with abstention mechanisms are evaluated using this benchmark.

**Strengths:**

1. The introduction of TrustSQL fills a critical gap in the evaluation of text-to-SQL models by incorporating infeasible questions.
2. The Reliability Score metric provides a meaningful way to quantify the trade-off between a model's helpfulness and harmfulness.
3. The paper provides an evaluation of different abstention mechanisms and their effects on model reliability.

**Weaknesses:**

1. Insufficient Comparison with Existing Work: The paper lacks a thorough and clear comparison with existing works that address similar challenges in handling infeasible questions in text-to-SQL tasks. Prior studies, such as TriageSQL and more recent research on hallucination problems in LLMs, have explored methods for detecting and managing unanswerable or infeasible queries. The paper does not adequately position its contributions within the context of these existing approaches. Moreover, it does not provide empirical comparisons with previous benchmarks and methods dealing with infeasible questions.
2. Lack of Quality Analysis for the Proposed Dataset: The paper does not provide quality metrics or detailed analyses to demonstrate the high quality of the re-annotated datasets and the newly added infeasible questions. Quality metrics such as inter-annotator agreement, dataset statistics, or validation processes are essential to establish the reliability and usefulness of the proposed benchmark. The author could also consider adding a user study or real-world deployment to validate whether the proposed RS metric aligns with actual user satisfaction or trust in the models.
3. Limited Evaluation Scope and Generalizability Concerns: The evaluation is limited to three simple domain-specific datasets, which may restrict the generalizability to other domains or more complex schemas found in cross-domain datasets like Spider and BIRD.

**Questions:**

1. The proposed classification of infeasible questions into missing-schema, ambiguous, and non-SQL categories may not be exhaustive. Are there other types of infeasible or unanswerable questions that the classification does not cover, such as questions based on inaccurate premises, malformed queries, or those that require external knowledge beyond the database schema? How would these additional categories impact model performance and evaluation? It would be helpful if the authors could discuss the potential for other classes of infeasible questions and how their benchmark could accommodate them.
2. Regarding the penalty c, is the selection and sensitivity of the c value thoroughly explored? Why choose 1, 0.5N, and N? Could the authors elaborate on how it was determined? Are there recommended strategies or guidelines for practitioners to select appropriate values in different application contexts?

---

> ### Author Response · Authors · 2024-11-27
> **Author Response**
>
> Thank you for taking the time to review our paper. Please find our responses below.
>
> **Q1: Comparison with existing work**
>
> Please refer to GC1 in the general comment section above for a detailed comparison table illustrating how TrustSQL differs from other benchmarks. In summary, TrustSQL uniquely integrates multiple types of input questions—including unanswerable, ambiguous, and answerable questions—and includes executable SQL with associated databases, features that many existing benchmarks lack.
>
>
> **Q2: Quality metric for dataset and annotation procedure**
>
> Since the dataset was not created through crowdsourcing, we did not report inter-annotator agreement statistics. Instead, the three authors of this paper, all proficient in SQL, served as annotators. We ensured high-quality annotations by resolving all disagreements until reaching consensus. The detailed annotation process is summarized in GC3 in the general comments section above.
>
> **Q3: Limited evaluation scope and generalizability**
>
> TrustSQL includes complex databases with an average of 18 tables per database and complex SQL queries involving multiple table joins (please refer to GC1 for more details). This complexity allows us to evaluate the reliability of text-to-SQL models more effectively. Regarding generalizability, the inclusion of the three domain-specific databases aims to evaluate whether methodological trends for reliable text-to-SQL modeling remain consistent across databases, rather than to assess a model's generalization ability in SQL generation across diverse databases. For further discussion, please refer to GC2 above.
>
> **Q4: Types of infeasible questions**
>
> As noted in our paper, it is not feasible to cover all possible types of infeasible questions in a benchmark dataset. Instead, we focus on the most problematic types identified in previous studies and conduct annotations based on these observations. We argue that if a model struggles with these types, it is unlikely to handle other question types effectively. While we considered adding more types of infeasible questions—including the categories you suggested—ambiguities in evaluation led us to select the current infeasible categories.
>
> Regarding the three classes you mentioned, our responses are as follows:
> - Inaccurate Premises: In text-to-SQL, feasibility is determined by whether the necessary information exists in the database schema, regardless of the factual accuracy of the premise. Therefore, the factual accuracy of a question is not critical in determining its feasibility.
> - Malformed Queries: Determining whether a question with typos or grammatical errors remains well-formed introduces ambiguity. It becomes challenging to decide if such a question should be classified as ambiguous or unanswerable. To maintain clear evaluation boundaries, we chose to exclude such cases but acknowledge this as an area for future work.
> - Requiring External Knowledge: To ensure a fair comparison across models with varying levels of world knowledge, we excluded questions requiring external knowledge. We focused on knowledge explicitly provided in the SQL assumptions for each database (Appendices A.1.1–A.1.3).
>
> **Q5: Recommended strategies for selecting c**
>
> We have elaborated on the selection of c in the general comments section above (see GC4). In summary, the choice of c depends on the safety requirements for model deployment and can vary based on user preferences, SQL proficiency, or organizational policies. We provide guidelines to assist users in selecting an appropriate c value for their specific needs.

---

### Official Review · Reviewer_Hikm · 2024-11-02

**Soundness:** 3
**Presentation:** 1
**Contribution:** 2
**Rating:** 5
**Confidence:** 3

**Summary:**

The paper introduces TrustSQL, a new benchmark designed to evaluate the reliability of text-to-SQL models. It addresses two significant gaps in existing benchmarks: the lack of infeasible questions that cannot be mapped to SQL queries and the absence of abstention mechanisms in current models. TrustSQL is constructed by re-annotating three datasets—ATIS, Advising, and EHRSQL—and incorporating infeasible questions to provide a comprehensive evaluation. The authors propose a novel metric called the Reliability Score (RS), which quantifies a model's helpfulness relative to its harmfulness, adjusted by a user-defined penalty. They evaluate text-to-SQL models integrated with various abstention mechanisms, such as classifiers and uncertainty estimation methods. The experiments reveal that most models fail to meet safety requirements under high penalty settings, underscoring the need for developing models that ensure a certain degree of reliability alongside SQL generation accuracy.

**Strengths:**

The paper makes a significant contribution by introducing TrustSQL, a benchmark that fills a critical gap in the evaluation of text-to-SQL models. By incorporating infeasible questions and proposing the Reliability Score (RS), the authors provide a fresh perspective on assessing model reliability, which is crucial for real-world deployment.
The authors conduct thorough experiments using both fine-tuning and in-context learning settings, integrating various abstention mechanisms. The meticulous re-annotation of existing datasets and the careful construction of infeasible questions enhance the quality and relevance of the benchmark. The paper's focus on reliability and safety has the potential to influence future research and practices in the field.

**Weaknesses:**

The paper evaluates several abstention mechanisms but could provide deeper insights into why certain methods perform better under specific conditions. A more thorough analysis of the trade-offs between coverage and risk for each mechanism would be beneficial.
The Reliability Score depends heavily on the user-defined penalty parameter. The paper does not offer sufficient guidance on how practitioners should choose this parameter in practice. A sensitivity analysis or guidelines would make the RS metric more practical.

**Questions:**

Can you provide practical advice or criteria for selecting the penalty parameter 'c' in the Reliability Score?
How sensitive is the RS to different values of 'c'?
Have you considered that frequent abstentions might affect user experience?

---

> ### Author Response · Authors · 2024-11-27
> **Author Response**
>
> Thank you for taking the time to review our paper. Please find our responses below.
>
> **Q1: Analysis of abstention mechanisms**
>
> Coverage and risk are valuable metrics for evaluating the correctness ratio of models; however, the Reliability Score (RS) offers a more comprehensive measure of reliability under a fixed penalty setting. Since our focus is on comparing model performance across varying penalty settings, coverage and risk alone do not provide sufficient insight for making informed model selection decisions.
>
> An important takeaway is that when the penalty for incorrect decisions is low, seemingly high-performing models like GPT-4o-powered baselines perform well because they maintain low risk while offering reasonable coverage. However, in scenarios with extremely high penalties, models with adjustable thresholds, such as T5-3B leveraging uncertainty estimation of the internal model state, are more suitable. This is because threshold adjustments enable finer control over decisions to abstain or answer, allowing the modeler to align the model’s behavior with specific safety requirements. In single-turn text-to-SQL scenarios where mistakes are unacceptable, setting stricter thresholds ensures that only SQL generation outputs with high certainty are considered—an essential feature for high-stakes applications.
>
> **Q2: Guidelines for selecting the penalty parameter c**
>
> We have elaborated on the selection of the penalty value c in the general comments section (see GC4). In summary, the choice of c depends on the safety requirements for model deployment, which can vary based on user preferences, SQL proficiency, or organizational policies. To assist users, we provide guidelines to help them select an appropriate c value tailored to their specific needs.

---

> > ### Comment · Reviewer_Hikm · 2024-11-27
> > **Thank you for your clarifications.**
> >
> > Thank you for your clarifications. The distinction between low and high penalty settings and the role of adjustable thresholds, such as in T5-3.8, is well explained. For selecting the penalty parameter  c, a more detailed, practical guideline would improve the usability of the Reliability Score. Additionally, discussing the user experience impact of frequent abstentions would strengthen the paper, particularly in high-stakes applications.

---

> > > ### Author Response · Authors · 2024-11-28
> > > **Author Response**
> > >
> > > Thank you for the response. Please note that the chosen c values in our work are not subjectively selected to favor our results (this is a benchmark study, not a method proposal). It would be interesting to consider the user experience impact of frequent abstentions, and we plan to address this in the final manuscript if possible.

---

### Official Review · Reviewer_iVir · 2024-11-05

**Soundness:** 2
**Presentation:** 2
**Contribution:** 3
**Rating:** 5
**Confidence:** 4

**Summary:**

The paper introduces TrustSQL, a benchmark designed to assess the reliability of text-to-SQL systems in handling both feasible and infeasible questions. To build the TrustSQL dataset, the authors re-annotated three existing datasets: ATIS, Advising, and EHRSQL. TrustSQL’s evaluation metric considers both question types, incorporating a novel metric, the Reliability Score. With this metric, correctly answering feasible questions and identifying infeasible questions yield a positive score of 1, while other responses result in a score of 0 or a negative penalty of -C.

The paper evaluates both classifier-based methods, which use a sub-model classifier to distinguish feasible and infeasible questions, and uncertainty-estimation-based methods, which rely on SQL generation uncertainty.

**Strengths:**

1. The paper introduces the TrustSQL dataset, aimed at evaluating text-to-SQL systems on both feasible and infeasible questions. The re-annotated datasets could serve as a valuable resource for the text-to-SQL community.

2. The paper evaluates multiple text-to-SQL systems, including the SOTA fine-tuned text-to-SQL model (SQLCoder) and a general-purpose LLM (GPT-4), using both classifier-based and uncertainty-estimation approaches for detecting infeasible questions.

**Weaknesses:**

1. Re-annotation is a central component of the paper and should be included in the main content rather than the appendix, especially given that one page of space remains available.

2. The authors find that many text-to-SQL models can produce responses that are potentially more harmful than helpful, but this observation depends on the choice of weight C in the penalty. The choices for C (set at 1, N/2, and N) seem arbitrary. The paper lacks a discussion on the rationale behind these values and whether any human studies informed this choice.

**Questions:**

1. Lines 200–201 mention that all questions were reviewed to ensure clarity, with adjustments made for SQL queries that were too similar. How was clarity ensured during annotation, and what metric was used to determine if a question was clear or if two SQL queries were overly similar?

2. Who are the annotators, and how are disagreements in annotation resolved?

---

> ### Author Response · Authors · 2024-11-27
> **Author Response**
>
> Thank you for taking the time to review our paper. Please find our responses below.
>
> **Q1: Re-annotation details should be included in the main content rather than the appendix**
>
> We agree that the re-annotation process is an important part of our paper and appreciate your suggestion. While we initially adhered to the 9-page recommendation from the Call for Papers website (https://iclr.cc/Conferences/2025/CallForPapers), we have now expanded the main content to 10 pages. Specifically, we have added detailed descriptions of the data annotation process in Section 4. This includes comprehensive information about the annotation methodology, the roles of the annotators, and the steps taken to ensure data quality.
>
> **Q2: The rationale behind the choice of c**
>
> We have elaborated on the selection of the penalty value c in the general comments section (see GC4). In summary, the choice of c depends on the safety requirements for model deployment and can vary based on user preferences, SQL proficiency, or organizational policies. We provide guidelines to assist users in selecting an appropriate c value for their specific needs.
>
> **Q3: Annotator details and resolving annotation inconsistencies**
>
> Three authors of this paper served as annotators, performing manual annotation and review to ensure the highest possible data quality (no crowdsourcing was conducted). During the review process, all annotators met in person to resolve annotation disagreements until consensus was reached. We have included common areas of annotation disagreement in the revised manuscript. For a brief summary of this process, please refer to GC3 in the general comments section above.
>
> **Q4: How was clarity ensured during annotation, and what metric was used to determine overly similar SQL?**
>
> We ensured clarity by verifying whether questions accurately reflect their corresponding SQL queries, without introducing implicit assumptions beyond the SQL assumption text (Appendix A.1.1–A.1.3). Each question-SQL pair was manually reviewed by the annotators at the sample level. To identify overly similar SQL queries, we first categorized question templates that use the same placeholders (e.g., "Tell me flights from city_name1 to city_name0" and "What are airlines that provide services from city_name1 to city_name0," which share the placeholders city_name1 and city_name0). Templates with the same placeholders were then checked to determine whether they have identical SQL conditions and logical structures. If they were found to be semantically identical, we merged the templates.

---

> > ### Comment · Reviewer_iVir · 2024-11-27
> >
> > Thank you to the authors for the clarification and for adding more details about the annotation process to the paper. However, I still have concerns about the choice of c. While the authors state that users can select the value, the study and conclusions in the paper are based on the authors’ chosen value. Without a user study, these conclusions may be subjective. Additionally, I am not fully convinced that BIRD relies solely on "sample-level" evidence and TrustSQL on "database-level" evidence. The assumption in the BIRD database appears to be shared across samples within each database, which should be suitable for the TrustSQL assumption.

---

> > > ### Author Response · Authors · 2024-11-28
> > > **Author Response**
> > >
> > > Thank you for the response. The penalties in our work are not subjectively chosen to favor our results, as this is a benchmark study rather than a method proposal. Instead, they merely represent distinct safety levels, which we believe are sufficient to demonstrate a proof-of-concept in an academic context. Even without a user study, using alternative c values (e.g., 10,000 instead of N) would not significantly affect the results or conclusions, as long as two extremes (1 and N) and a middle value (10, N/2, or another value in between) are included. The key takeaway remains unchanged: for applications with higher penalty requirements, current high-performing models still exhibit notable shortcomings, highlighting the need for targeted improvements.
> > >
> > > Regarding the assumptions in BIRD, let us consider the following examples:
> > >
> > > {'db_id': 'movie_platform',
> > >  'question': 'What is the name of the longest movie title? When was it released?',
> > >  'evidence': 'longest movie title refers to MAX(LENGTH(movie_title)); when it was released refers to movie_release_year;',
> > >  'SQL': 'SELECT movie_title, movie_release_year FROM movies ORDER BY LENGTH(movie_popularity) DESC LIMIT 1'}
> > >
> > > {'db_id': 'movie_platform',
> > >  'question': 'Name the movie with the most ratings.',
> > >  'evidence': 'movie with the most rating refers to MAX(SUM(rating_score));',
> > >  'SQL': 'SELECT movie_title FROM movies GROUP BY movie_title ORDER BY COUNT(movie_title) DESC LIMIT 1'}
> > >
> > > The above are two sample data points from BIRD. As shown, the evidence is specific to each example. In contrast, the SQL assumptions in A.1.1–A.1.3 are defined at the database level, removing the assumption that all input questions are feasible while still enabling the text-to-SQL task. The use of evidence is left to the model's discretion.

---

> > > > ### Comment · Reviewer_iVir · 2024-11-29
> > > >
> > > > Thank you for the prompt response. I noticed that other reviewers also share concerns about the choice of c. I believe that harmfulness and helpfulness should be considered as two separate dimensions to measure these two things clearly. Additionally, while each example is provided with oracle knowledge, there is no conflict in merging these examples together as part of the database evidence.

---

> > > > > ### Author Response · Authors · 2024-12-01
> > > > > **Author Response**
> > > > >
> > > > > Thank you for engaging in the discussion with us. Please find our response below.
> > > > >
> > > > > **Q1: Reason for measuring both harmfulness and helpfulness simultaneously**
> > > > >
> > > > > In the final phase of text-to-SQL development (i.e., real-world deployment), distinguishing between helpfulness and harmfulness becomes impractical. For example, medical alert systems (e.g., mortality prediction) use metrics like AUROC (i.e., tradeoff between recall and false alarm rate) or Precision at Recall of k (i.e., precision with some recall guarantee). These metrics inherently capture both helpfulness (i.e., How accurate is the prediction?) and harmfulness (i.e., How much time/resource does it waste due to inaccurate predictions?). Similarly, when text-to-SQL serves as an end-product that provides answers to user questions, we believe it is essential to evaluate it by considering the tradeoff between helpfulness and harmfulness. Since the degree of harmfulness perceived by end-users is subjective and cannot be directly measured, we introduce it as a variable, c.
> > > > >
> > > > > **Q2: Exclusion of BIRD**
> > > > >
> > > > > We can combine the evidence if we disregard naturalness, but we believe these sample-level assumptions are not shared across samples, making it difficult to consider this as database-level evidence. More importantly, as stated in our paper, the inclusion criteria we prioritize are highly complex text-to-SQL databases, which only a small portion of databases in BIRD satisfy (8 databases with ≥15 #Tab/DB). Additionally, BIRD's template-free annotations complicate quality control, whereas domain-specific datasets' template-based annotations provide better control over the semantic distribution of questions across data splits. Since our experiments are not conducted in cross-database settings (we use trainable in-domain question-SQL data for fine-tuning and in-context learning for SQL generation), we argue that adding a few more databases from BIRD with the considerable effort of converting them into single-database settings (e.g., templatization, paraphrasing, value sampling, etc.) would not significantly alter the core claims or results of our work.

---

### Author Response · Authors · 2024-11-27
**General Comment 2**

**GC2. Database selection (Why not using Spider or BIRD)**

Due to the nature of the task and the penalty-based scoring—which amplifies the impact of incorrect model decisions—we ruled out large-scale, template-free SQL annotations obtained through crowdsourcing. Instead, we selected a collection of three highly complex, domain-specific datasets (with complexity reported in GC1 above) that reflect diverse real-user questions. We manually re-annotated them to ensure both high-quality annotations and task complexity.

The inclusion of these three domain-specific databases is intended to provide a setting where the methodological trends for reliable text-to-SQL modeling remain consistent across databases (no cross-domain experiments are conducted in this work), rather than to assess a model's cross-domain generalization ability in SQL generation—which is the focus of other cross-domain datasets like Spider and BIRD.

We also considered re-annotating Spider and BIRD to expand database coverage but found limitations. Spider is mostly solved using GPT-4o, with remaining errors due to annotation issues rather than SQL generation challenges. BIRD relies on "sample-level" evidence, incompatible with TrustSQL's setup where not all input questions are text-to-SQL feasible. TrustSQL uses "database-level" evidence (i.e., SQL assumption text in Appendix A.1.1–A.1.3) shared across samples, removing the assumption that all input questions are feasible while still enabling the text-to-SQL task by using evidence only when necessary.


**GC3. Details on annotators and resolving annotation inconsistencies**

To ensure high-quality annotations, the three authors (all proficient in SQL) served as annotators without relying on crowdsourcing. Below, we describe their roles in each stage of the annotation process.

- Feasible question annotation process (template-based)
  - Review and modification: One annotator reviews and re-annotates the question templates and their corresponding SQL structures. Template merging occurs if the templates are semantically identical (approx. 550 templates across datasets).
  - Paraphrase generation: Two annotators generate new natural language questions for each question template using GPT-4o and review them to ensure they coherently reflect the original question template.
  - Pair construction: One annotator merges the paraphrases with the corresponding question templates.
  - Review: All annotators engage in real-time discussions to resolve any disagreements in annotation until they reach consensus.

- Infeasible question annotation process (keyword-based)
  - Keyword-guided annotation: Annotators are given specific keywords for annotating infeasible questions (e.g., hypothetical columns for missing-schema questions, ambiguous terms for ambiguous questions, and task names for non-SQL questions), along with sample feasible questions. They are tasked to manually modify the feasible questions using the provided infeasible keywords to make them infeasible—questions that resemble feasible ones but are, in fact, infeasible.
  - Review: The annotated questions are reviewed in real-time until all disagreements among the authors are resolved, ensuring consistency and high-quality annotations.


**GC4. Interpreting the Reliability Score and the choice of the penalty**

The choice of the penalty value is user-defined and depends on the safety requirements for model deployment. These requirements can vary based on user preferences, SQL proficiency, or organizational policies. While determining the best c is beyond this work's scope, we offer meaningful penalty scenarios:
- Lenient scenario (c = 1): A single mistake carries the same weight as one correct model decision. A positive RS here indicates the model makes more correct decisions than incorrect ones.
- Moderate scenario (c = 10): Every 10 correct decisions carry the same weight as one incorrect decision, reflecting a moderate tolerance for errors.
- Strict scenario (c = N/2): Two mistakes result in a negative RS, even if all other decisions (N – 2 out of N total cases) are correct.
- Most strict scenario (c = N): A single mistake results in a negative RS, even if all other decisions (N – 1 cases) are correct. Models must avoid any mistakes to achieve a positive RS.

Once the penalty value c is set, the best model can be selected based on its performance in the RS. Below are the key guidelines for model selection using the RS:
- Model comparison: A model's superior performance under specific penalties does not imply it outperforms others across different penalty settings. Models should be compared within the same penalty setting.
- Interpreting the score: Models with positive RS values are preferable for deployment, as they meet safety requirements and demonstrate greater helpfulness than harmfulness. A model that ranks the highest but achieves a negative RS should be reconsidered for deployment.

---

### Author Response · Authors · 2024-11-27
**General Comment 1**

We thank all the reviewers for their valuable comments and suggestions. We have revised our manuscript based on the feedback provided. Common questions raised by the reviewers are addressed in this general comment (GC) section, while responses to specific points raised by each reviewer are organized point by point in their respective sections.

**GC1. Comparison with other benchmarks**

| Dataset          | UnansQ | AmbigQ | AnserQ | Exec SQL/DB^ | Evidence§ | #DB | #Tab/DB | #Tab/SQL‡ | Avg SQL Tok | Eval Penalty* |
|----------------|------------------------|--------------------|---------------------|-------------------------|------------|------|-------------|-----------------|--------------------|-----------------|
| TriageSQL      | ✓                      | ✓                  | ✓                   | ✕                       | ✕          | -†    | 1.7         | -               | -                  | ✕               |
| DTE            | ✓                      | ✓                  | ✓                   | ✕                       | ✕          | -    | 1.0         | -               | -                  | ✕               |
| WikiSQL        | ✕                      | ✕                  | ✓                   | ✓                       | ✕          | -    | 1.0         | 1.0             | 12.1               | ✕               |
| Spider         | ✕                      | ✕                  | ✓                   | ✓                       | ✕          | 200  | 5.1         | 1.6             | 18.6               | ✕               |
| KaggleDBQA     | ✕                      | ✕                  | ✓                   | ✓                       | ✕          | 8    | 2.3         | 1.2             | 17.3               | ✕               |
| EHRSQL         | ✓                      | ✕                  | ✓                   | ✓                       | ✕          | 1    | 17.0        | 2.4             | 68.9               | ✕               |
| BIRD           | ✕                      | ✕                  | ✓                   | ✓                       | ✓          | 95   | 7.5         | 2.0             | 31.1               | ✕               |
| TrustSQL       (this work)              | ✓                  | ✓                   | ✓                       | ✓          | ✓    | 3          | 18.0            | 2.9               | 60.5            | ✓               |

^ Indicates whether the dataset contains question-SQL pairs and corresponding databases that can produce execution results.
§ Indicates whether the dataset contains textual hints/knowledge to guide SQL generation.
† Indicates that the dataset contains string-formatted database schemas specific to each sample (TriageSQL) or single-table settings (DTE and WikiSQL).
‡ Indicates the number of unique tables used for each SQL query.
\* Indicates whether model mistakes are penalized during evaluation.

TrustSQL is unique in integrating multiple types of input questions for SQL generation, including both feasible (answerable) and infeasible (unanswerable and ambiguous) questions. Existing benchmarks like TriageSQL and DTE include all these types of questions, enabling question classification prior to SQL generation. However, they lack executable SQL and corresponding databases, limiting their scope in addressing errors related to SQL generation. Meanwhile, most standard text-to-SQL benchmarks focus solely on SQL generation under the assumption that all input questions are feasible, leaving the challenge of handling infeasible questions unaddressed. Additionally, they do not consider abstention when the SQL is likely to be incorrect, as their evaluation metrics (Eval penalty in the table above) do not penalize incorrect answers. From the end-user's perspective, reliability is paramount—they need text-to-SQL models that provide correct responses (helpful) while avoiding incorrect ones (harmful) by abstaining when necessary. TrustSQL is designed to directly address this need by evaluating models not only on their ability to generate SQL but also on their reliability and decision-making in the face of uncertainty.

---

### Note · Authors · 2024-12-03

**Comment:**

We thank the reviewers for their time and feedback on our paper. After careful consideration, we have decided to withdraw our submission to further develop the work in alignment with the feedback provided.

**Withdrawal Confirmation:**

I have read and agree with the venue's withdrawal policy on behalf of myself and my co-authors.